# Manganese co-limitation of phytoplankton growth and major nutrient drawdown in the Southern Ocean

Thomas J. Browning [1]✉, Eric P. Achterberg [1], Anja Engel[1] & Edward Mawji [2]

Residual macronutrients in the surface Southern Ocean result from restricted biological utilization, caused by low wintertime irradiance, cold temperatures, and insufficient micronutrients. Variability in utilization alters oceanic $CO_2$ sequestration at glacial-interglacial timescales. The role for insufficient iron has been examined in detail, but manganese also has an essential function in photosynthesis and dissolved concentrations in the Southern Ocean can be strongly depleted. However, clear evidence for or against manganese limitation in this system is lacking. Here we present results from ten experiments distributed across Drake Passage. We found manganese (co-)limited phytoplankton growth and macronutrient consumption in central Drake Passage, whilst iron limitation was widespread nearer the South American and Antarctic continental shelves. Spatial patterns were reconciled with the different rates and timescales for removal of each element from seawater. Our results suggest an important role for manganese in modelling Southern Ocean productivity and understanding major nutrient drawdown in glacial periods.

[1] Marine Biogeochemistry Division, GEOMAR Helmholtz Centre for Ocean Research, Kiel, Germany. [2] National Oceanography Centre Southampton, Southampton, UK. ✉email: tbrowning@geomar.de

All of Earth's photosynthetic organisms have an essential manganese (Mn) requirement for the water-splitting reaction in oxygenic photosynthesis[1]. Such a requirement leads to relatively restricted flexibility in phytoplankton Mn quotas, which are ~2.7 times smaller than for iron (Fe)[2]. Throughout much of the global ocean, dissolved Mn concentrations are characterized by surface ocean maxima and depletion in the ocean interior (typically <0.2 nM)[3,4]. This trend contrasts with most other micronutrients, including Fe, and results from a combination of (i) input of Mn-bearing aerosols into stratified surface waters[5], and (ii) much higher reduction rates of insoluble $Mn^{3+/4+}$ oxides to soluble $Mn^{2+}$ in sunlight[6,7], dissolving Mn oxides and buffering against oxidation of $Mn^{2+}$ that proceeds at all depths[8].

The Southern Ocean contrasts with much of the global ocean in a combination of four ways pertinent to the availability of surface dissolved Mn: (i) dust inputs are very low[9]; (ii) surface waters are fed by strong upwelling of low Mn waters from the oceans interior[4,10]; (iii) low surface light levels prevail for a large part of the year[11], which, potentially in combination with lower dissolved organic matter reducing agents[6,12], restrict Mn photoreduction[6,7]; and, (iv) concentrations of major nutrients (nitrate and phosphate) are perennially high and do not limit biological micronutrient (e.g. Fe, Mn) drawdown[2]. As a result, measurements in surface waters of the Southern Ocean have demonstrated that Mn, alongside Fe, can be strongly depleted[13–18].

More complete utilization of upwelled major nutrients in the Southern Ocean before they are subducted again reduces ocean-to-atmosphere $CO_2$ transfer[19]. Variability in this process is one of several mechanisms expected to contribute to the ~90 p.p.m. glacial-to-interglacial atmospheric $CO_2$ changes recorded in ice cores[20]. At present, low Fe availability is thought to limit phytoplankton growth in the modern-day Southern Ocean[2,11]. By inference, Fe fertilization by higher atmospheric dust fluxes[21,22] is considered a key driver of major nutrient utilization and reduced atmospheric $CO_2$ in glacial periods[23–26]. Differences in the biogeochemical cycles of Fe and Mn[6,7,27–32], imply that the occurrence of, and controls on, Mn rather than Fe limitation in the Southern Ocean would therefore be important for a mechanistic understanding of glacial-interglacial major nutrient drawdown and $CO_2$ changes[23–26] as well as predicting Earth System feedbacks to future climate conditions[11,33].

Despite this, few experimental tests have been conducted to assess the relative role of Fe and/or Mn in regulating phytoplankton growth in the Southern Ocean[18,32,34–36]. To confront this, we conducted ten trace-metal-clean bioassay incubation experiments in austral spring on a cruise track spanning Drake Passage (Fig. 1). Factorial combinations of Fe and Mn were supplied in triplicate, and net phytoplankton accumulation, photophysiology and community composition changes, and dissolved macronutrient drawdown were assessed after periods of 2–5 days relative to untreated controls (Supplementary Table 1). We found that Mn was (co-)limiting to phytoplankton growth and major nutrient drawdown in central Drake Passage, whereas Fe was limiting nearer the South American and Antarctic continental shelves. We explain these trends by the different removal kinetics of Fe and Mn in seawater and their operational timescales within the different waters entrained into the surface Southern Ocean. Finally, simulations with a simple ecosystem model suggest that enhanced dust deposition, for example during glacial periods, could lead to greater limitation by Mn rather than Fe in the Southern Ocean.

## Results and discussion

**Phytoplankton responses to combinations of Fe and Mn supply.** Phytoplankton responses to combinations of Fe and/or

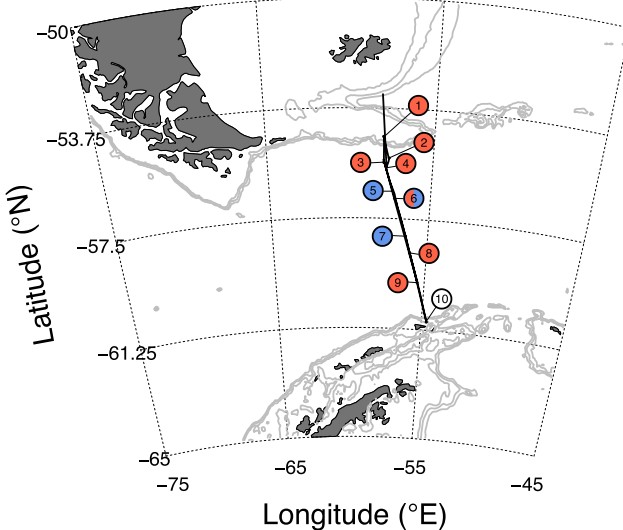

**Fig. 1 Experiment locations through Drake Passage.** Red symbols indicate Fe-limited sites (as indicated by chlorophyll-a responses; Fig. 2); Blue symbols indicate Mn-limited sites. Split red-blue symbol indicates Fe–Mn co-limitation. White symbol indicates no nutrient limitation detected. Thick back line indicates the cruise transect. Gray lines indicate 0.5, 1, and 1.5 km bathymetric depth contours.

Mn supply showed a clear cross-passage gradient (Figs. 1 and 2). At the edges of Drake Passage, which were proximal or downstream of shallower continental shelves, phytoplankton concentrations (approximated by chlorophyll-a) were either (i) Fe-limited and exhibited no serial (i.e., supplementary) responses to Fe+Mn supplied in combination (Fig. 2a–d, h, i), or (ii) in the one experiment initiated 15 km from Elephant Island (Antarctic Peninsula), not limited by any nutrient (as evidenced by the large increase in phytoplankton in all experimental bottles, including untreated controls; Fig. 2j). Conversely, in the core of Drake Passage and most distal to continental shelves, experiments showed either (i) primary Mn limitation followed by serial Fe limitation (i.e., the addition of Fe+Mn led to more growth than supply of Mn alone) (Fig. 2e, g), or (ii) co-limitation by both Mn and Fe (i.e., both nutrients were needed to stimulate a growth response) (Fig. 2f).

For the experiments classified as primary Mn limited (Fig. 2e, g), small but significant increases in chlorophyll-a within +Fe treatments were also observed. This implied some degree of Mn-Fe co-limitation at these sites, which could have been caused by either, or a combination of, (i) stimulation of different communities, each limited by one of the nutrients (either Mn or Fe), and/or (ii) substitutability of Fe and Mn in superoxide dismutase enzymes[31], meaning that supply of either nutrient would reduce requirement for the other[32,37]. Bulk particulate organic carbon measurements of pooled replicate treatments generally followed chlorophyll-a trends, for instance, they demonstrated enhancements in +Mn over +Fe treatments in Experiments 5 and 7 (Fig. 2l; Supplementary Fig. 1). Dissolved macronutrient drawdown during incubations matched biomass increases, with Mn additions at the two Mn-limited sites producing significantly enhanced macronutrient drawdown as a result of biomass formation relative to either untreated controls or +Fe amendment (Fig. 2n, o; Supplementary Fig. 1).

Supplying limiting nutrients stimulated phytoplankton groups initially present. Diatoms dominated initial phytoplankton biomass throughout the region (Supplementary Fig. 2), and concentrations of fucoxanthin—a pigment associated with diatoms—increased most following addition of nutrient(s)

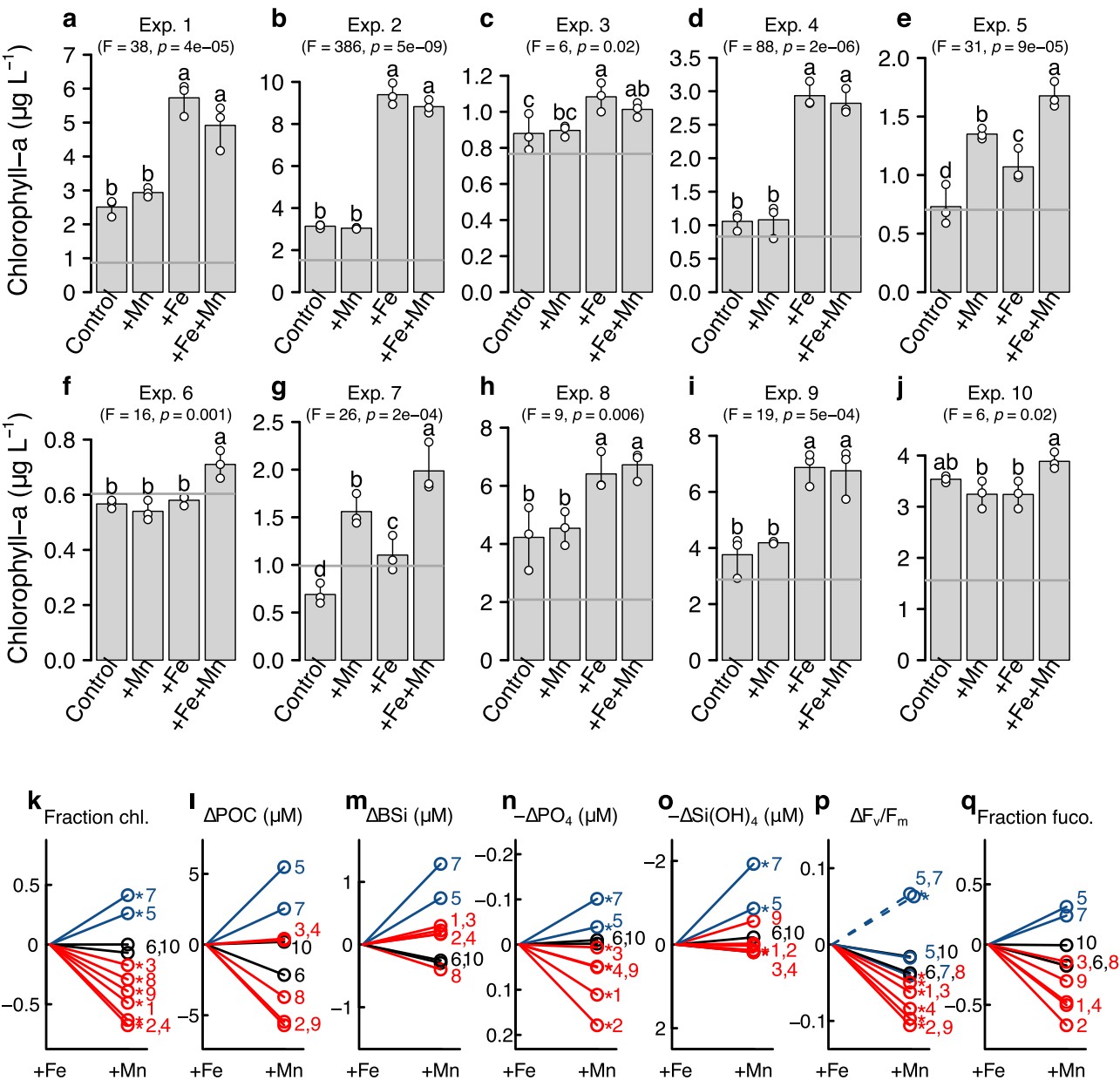

**Fig. 2 Biochemical responses to Fe and Mn supply. a–j** Chlorophyll-a changes in Experiments 1–10 (Experiment 1=North Drake Passage; Experiment 10=South Drake Passage; Fig. 1). Bar heights indicate the mean of triplicate replicates (white symbols; line is the range). For each experiment, different letters above bars indicate statistically different mean responses (one-way ANOVA, $F$ and $p$ values shown, followed by Fisher least significant difference test). Arrows indicate the average initial chlorophyll-a ($n = 3$). **k–q** Comparison of biochemical and physiological changes following supply of Fe or Mn. A flat line indicates no difference in responses between Fe and Mn additions, negative slopes indicate greater responses to Fe addition, and positive slopes indicate greater responses to Mn addition. Fe-limited sites are shown in red and Mn-limited sites in blue, as determined from the chlorophyll-a responses in **a–j**. The nutrient replete site (Exp. 10) and the Fe–Mn co-limited site (Exp. 6) are in black. Numbers indicate the experimental sites. Parameters shown are: fractional chlorophyll-a change between Fe and Mn treatments (**k**), differences in POC and BSi accumulation (**l, m**), differences in phosphate and silicic acid drawdown (**n, o**), differences in $F_v/F_m$ (**p**), and fractional fucoxanthin changes between Fe and Mn treatments (**q**). Dashed lines in **p** show $F_v/F_m$ responses of the serially Mn-Fe-limited sites (Exps 5 and 7) to combined Fe+Mn supply. Where parameters had triplicate biological replicates (all apart from POC, BSi, and fucoxanthin), stars indicate differences in parameter means between +Fe and +Mn were statistically significant (one-way ANOVA $p < 0.05$, followed by Fisher Least Significant Difference test).

limiting to overall phytoplankton biomass (i.e., +Fe at Fe-limited sites and +Mn at Mn-limited sites; Fig. 2r; Supplementary Fig. 2). Changes in the concentrations of other pigments and cell counts of smaller phytoplankton, resolved by flow cytometry analyses, were less clear or insignificant following nutrient supply at either Fe or Mn-limited sites (Supplementary Fig. 3).

Changes in phytoplankton photophysiology were consistent with limitation patterns identified by biomass changes and macronutrient drawdown. Fe supply alone enhanced the apparent photosystem II photochemical efficiency parameter, $F_v/F_m$, in waters nearer the edges of Drake Passage. At the Mn(-Fe) (co-) limited sites in central Drake Passage $F_v/F_m$ showed small

changes following either Fe or Mn supply, but larger increases in response to supply of Fe+Mn (Fig. 2p; Supplementary Fig. 1). More restricted increases in $F_v/F_m$ following recovery from Mn limitation in +Mn treatments likely reflected phytoplankton transitioning from Mn limitation into Fe limitation following Mn supply. The latter transitions were potentially linked with changes in residual silicic acid: nitrate concentration ratios in the experiments (Supplementary Fig. 4). As previously observed for Fe amendments under Fe-limited conditions[38,39], residual silicic acid: nitrate ratios increased over controls following Fe addition in Experiments 2, 4, and 9 (changes in Experiments 1 and 3 were insignificant), resulting from reduced silicic acid utilized per unit phytoplankton biomass produced. Conversely, in Mn-limited Experiments 5 and 7, residual silicic acid: nitrate ratios decreased following Mn addition relative to controls and +Fe treatments. This was despite calculated diatom contributions showing little or no differences between treatments (contributions to chlorophyll-a of 78–79% (Fe–Mn) and 84% (Fe and Mn) in Experiments 5 and 7, respectively). Whilst this observation in itself could be readily explained by Mn supply at these sites rapidly switching the main limiting nutrient from Mn to Fe, and thereby driving Fe-stress-induced increases silicic acid utilized per biomass formation[38,39], the +Fe+Mn treatments also showed reduced silicic acid: nitrate ratios. This suggested that transitions from Mn limited to either Fe-limited or (micro)nutrient replete conditions led to increased silicic acid: nitrate drawdown. In summary, the prevalence of Mn versus Fe limitation appeared important for the relative drawdown of Southern Ocean silicic acid and nitrate following micronutrient supply.

**Predictability of Fe versus Mn limitation by Mn\*.** The springtime dissolved Fe (DFe) and dissolved Mn (DMn) concentrations in central Drake Passage were slightly elevated with respect to previous observations, which were all made in mid-late summer[13,16,18,40] (Supplementary Table 1; cf. central Drake Passage GEOTRACES observations of DFe = 0.049–0.37 nM[40] and DMn = 0.086–0.2 nM[16]). Chlorophyll-a concentrations were also elevated with respect to typical later season values[18,41]. We attribute these differences to continued biological drawdown of both Fe and Mn through the growth season, with biological demand (approximated by phytoplankton biomass) tracking the availability of these nutrients.

Together, DFe and DMn concentrations at the experimental sites were a good predictor of phytoplankton responses to supplied nutrient combinations. Defining the deficiency of DMn relative to DFe for phytoplankton growth as Mn\* = DMn–DFe/$R_{Fe:Mn}$, where $R_{Fe:Mn}$ represents the assumed-average Fe:Mn ratio of phytoplankton (2.67)[2], showed elevated Mn\* at the Fe-limited sites towards the edges of Drake Passage (0.16–0.31 nM) and depressed Mn\* at Mn limited (or Mn-Fe co-limited) sites in central Drake Passage (–0.02 to 0.04 nM). Mn\* was also elevated at the one site located in close proximity to Elephant Island (2.35 nM), where concentrations of both DFe and DMn were high (>2 nM) and phytoplankton were not limited by either nutrient (Fig. 2j). Under the latter conditions, Mn\* relates to which of either Fe or Mn would be expected to become limiting following continued biological drawdown[42].

Overall, cross-Drake Passage trends in Mn\* measured in this study were consistent with earlier surveys in this region and other meridional sections elsewhere in the Southern Ocean[13,14,16–18,40,43,44] in showing depressed surface values close to 0 nM in open Southern Ocean surface waters (Supplementary Fig. 5). Provided the assumed-average Fe:Mn ratio of phytoplankton, $R_{Fe:Mn}$, remains relatively conserved across the spatial-temporal timescales of the observations, our experimental results suggest that these low Mn\*

values also reflect widespread Mn limitation, or Fe–Mn co-limitation, throughout open Southern Ocean regions. Such a finding is also consistent with observations of deep-water Mn\*. A key feature revealed in recent global-scale oceanic trace element surveys of the GEOTRACES program has been widespread depletion of DMn throughout the deep ocean, to concentrations that are typically more biologically deficient than DFe (i.e., negative Mn\*)[2]. As waters originating from the deep (<1 km) interiors of the three southern hemisphere oceans upwell and spiral east throughout the Southern Ocean[10], retention of low or negative deep-water Mn\* in surface waters and corresponding observations of phytoplankton Mn limitation should perhaps not be surprising. An emerging question however is how to reconcile negative deep-water Mn\* with the frequent occurrence of Fe rather than Mn limitation in parts of Drake Passage and other regions of the Southern Ocean[2,45] (Figs. 1, 2 and Supplementary Fig. 5)?

**Drivers of Fe versus Mn limitation in the Southern Ocean.** Although Fe and Mn have similar sources to the ocean, removal mechanisms differ[4,30]. The observed spatial patterns of surface Fe versus Mn limitation can therefore be understood in terms of the different removal kinetics of the two elements and how they operate on variable timescales (Fig. 3; See Methods). To demonstrate this, we paired equations describing the removal of DFe and DMn from seawater, in order to simulate changes in each nutrient and Mn\* over time (see Methods for equations and further details). Removal of DFe by scavenging processes is highly sensitive to DFe concentrations relative to those of stabilizing Fe-binding ligands (L), with faster DFe removal at DFe > L than at DFe < L[46]. Removal of DMn, on the other hand, is a result of the net balance between the oxidation of DMn to solid Mn oxides, which then sink, and the reduction of Mn oxides back to DMn[4,7].

We initiated simulations with subsurface DFe-DMn concentrations pairs from a published observational data set from around the Antarctic Peninsula[15,47], which reflect waters in close contact with shelf micronutrient sources (circles in Fig. 3a; Methods). In contrast to the deep open ocean, these subsurface (>200 m depth) near-shelf waters have high concentrations of both DFe and DMn, and Mn\* is elevated (e.g., >92% DFe-DMn concentrations pairs have Mn\* > 0.2 nM; $n$ = 26). The simulations show that this elevated Mn\* is retained for multiple years, as a result of relatively fast DFe removal at high concentrations in comparison with DMn. Sensitivity experiments showed that this extended period of Fe deficiency relative to Mn (i.e., positive Mn\*) was robust to a range of Mn oxidation and sinking rate constants (Methods). This analysis illustrates that that extended isolation time periods of such waters from further shelf inputs are required before Mn can approach levels of Mn-Fe co-deficiency (i.e., Mn\* ~0 nM). Although exact timescales for co-deficiency to be reached are sensitive to parameterization choices in the simulations, they are nevertheless robustly much longer than the days–weeks expected for off-shelf transports in the region, which are driven by entrainment of shelf waters by the eastward flowing Antarctic Circumpolar Current (Methods)[48,49]. High Mn\* signatures would therefore be maintained as these waters are advected away from the shelf and mixed vertically into the surface mixed layer[48,49].

In contrast to near-shelf waters, deep-water column GEO-TRACES measurements made in central Drake Passage and further east in a remote sector of Atlantic Southern Ocean (crosses in Fig. 3a)[16,40,43,44], have low Mn\* values, owing to longer time periods in the oceans interior away from terrestrial sources (e.g., >96% DFe-DMn concentrations pairs have Mn\* < 0.2 nM; $n$ = 30; shown as red ticks in Fig. 2c). Extending the previously described simulations for longer time periods

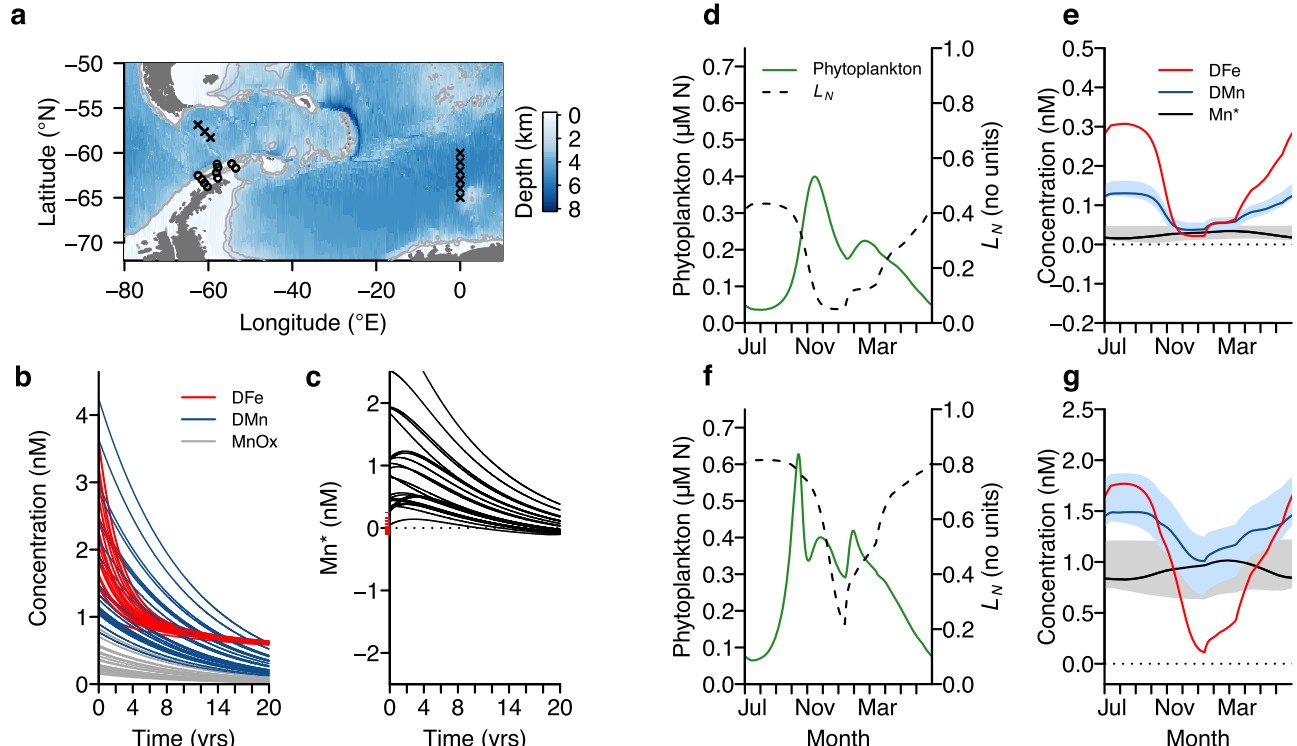

**Fig. 3 Controls on Fe versus Mn limitation in the Southern Ocean. a–c** Evolution of Mn* in deep waters. Waters in close contact with shelf sediments (circle locations in **a**) have elevated Mn* (concentrations indicated as $t = 0$ values in **b** and **c**; Methods)[15,47], which persists when scavenging removal is modeled for each element over the simulation timescale (lines in **b** and **c**; Methods). In contrast, deep open ocean waters isolated from sediments (cross locations in **a**) have low Mn* (red ticks in **c**). **d–g** Ecosystem model simulation of additional surface ocean processes. Deep-water DFe and DMn concentrations in the model were set to: **d–e**, mean observations for open ocean (crosses in **a**), and **f–g**, shelf-impacted waters (circles in **a**). $L_N$ is phytoplankton nutrient limitation in the model, where lower values indicate stronger nutrient limitation. Shading represents the model ranges generated when the rate constant for Mn reduction is set to either dark or high light values; solid lines are the results when the rate constant is scaled linearly between dark and light values as a function of mean mixed layer irradiance (see main text and Methods).

eventually leads to all simulated Mn* values for each Fe–Mn concentration pair tending towards these low, deep open ocean Mn* observations, however the exact Mn* concentration that is reached is sensitive to important unresolved mechanisms that must prevent DMn from descending to 0 nM. The low Mn* of these isolated deep waters prime them for phytoplankton Mn limitation, or Mn-Fe co-limitation, upon upwelling to the surface ocean[10], provided they do not interact substantially with high Mn* shelf-impacted waters previously described.

Once entrained into surface mixed layers, the difference in removal rate between DFe and DMn in both scenarios is extended by photoreduction of Mn, which can speed up the conversion of Mn oxides to free Mn ions by >50 times compared with rates in the dark[6,7]. This both protects DMn from removal and dissolves Mn oxides entrained from deep waters[6,7] and is a key mechanism maintaining elevated DMn in surface waters of the lower latitude oceans[3,4,8]. In addition, in sunlit surface waters biological utilization of DFe and DMn will interact with concentration-dependant removal processes (DFe scavenging, DMn oxidation). To investigate the impact of these additional dynamics in a Southern Ocean setting, we modified a simple ecosystem model[50] to include Fe and Mn as nutrients. We parameterized the deep waters in the model, which supply surface waters with nutrients during seasonal mixed layer deepening, with the mean average (i) near shelf, and (ii) open ocean subsurface DFe and DMn concentrations from the previously described observational datasets (Fig. 3a, d–g; Methods). Within this framework, we then adjusted the rate constant for Mn reduction to either a dark value (0.040 d$^{-1}$)[3,4,7], a bright light value (2.35 d$^{-1}$)[7], or a dynamic value

scaled linearly between these two extremes using mean mixed layer light in the model. In contrast to the strong expected control at lower latitudes, we found that photoreduction-related changes in DMn availability (individually and as Mn*) were secondary in comparison with the larger seasonal increases driven by wintertime deep mixing entrainment, and decreases in spring–summer resulting from phytoplankton growth (shading in Fig. 3d–g; Methods). As a result, values of surface ocean Mn* in the model remained elevated (Fe deficient) in the near-shelf case and close to 0 nM (Mn-Fe co-deficient) in the open ocean case, reflecting the sub-mixed layer Fe and Mn supply stoichiometry. Spring–summer phytoplankton growth in the model, therefore, becomes Fe–Mn co-limited in the open ocean case (Fig. 3d, e) and Fe-limited in the near-shelf case (Fig. 3f, g), matching our bioassay experiment results (Figs. 1 and 2).

**Implications of Southern Ocean Mn limitation.** Our finding of Mn limitation is important for understanding the drivers of productivity and associated feedbacks in the Southern Ocean[11,25]. Most immediately, Mn (co-)limitation throughout the extensive Mn deficient regions of the Southern Ocean (Supplementary Fig. 5) would restrict the fertilization impact of any mechanism exclusively enhancing the supply of Fe to these regions. This could include mechanisms such as anthropogenic Fe supply[45] or increased Fe ligand concentrations[51]. Furthermore, as current ocean-climate models do not include Mn this would also be an important factor, among others[52,53], in restricting the accuracy of their predicted responses to such forcing in these regions.

It is possible that Southern Ocean Mn limitation was more prevalent in the past. In glacial periods, enhanced dust Fe is thought to have increased diatom growth[54] and hence nitrate utilization[55], potentially driving a major fraction of the atmospheric $CO_2$ drawdown recorded in ice cores[24–26]. Typical continental-derived dust Fe:Mn ratios exceed assumed-average phytoplankton requirements >20-fold[2,27], implying that, with other factors staying the same, greater dust deposition might strengthen Mn limitation, in addition to stimulating overall increases in productivity and major nutrient drawdown. Reported solubilities of Mn and Fe in dust are wide-ranging: a recent compilation of southern South Atlantic and Southern Ocean solubilities found interquartile ranges of 3.1–13.5% for Fe ($n = 19$) and 6.5–39.2% for Mn ($n = 25$)[28]. Calculations with the extremes of these solubility ranges, applied to typical continental crust concentrations[27], indicate a Mn deficient dust source in each case (range of 1.7–44.7 fold higher soluble Fe:Mn than assumed-average phytoplankton requirements). In the ocean, the impact of this dust flux on Mn* would be modulated by Fe scavenging, Mn redox processes, and mixing with deep waters. If we repeat our simple ecosystem model simulation for the open ocean scenario (deep-water Mn* = 0.015 nM; Fig. 3e), but include elevated glacial dust deposition rates with the observed solubility ranges, we find that Mn limitation intensifies (Mn* becomes negative) under all scenarios relative to low-level modern-day dust deposition (Supplementary Fig. 6).

A more Mn limited glacial Southern Ocean could potentially be linked to the observed changes in residual silicic acid: nitrate concentration ratios in our experiments. We found that Fe supply at Fe-limited sites generally enhanced residual concentration ratios, matching earlier observations in Fe-limited systems[38,39]. In contrast, ratios decreased following Mn supply at the Mn limited sites (Supplementary Fig. 4). This indicated that conditions of either Mn or Fe limitation are an important control on the relative drawdown of nitrate and silicic acid. If this is the case, this is pertinent because whilst geochemical proxies suggest enhanced Southern Ocean nitrate utilization[55], they also indicate silicic acid was not drawdown[56], which is at odds with increased diatom growth[54]. If Mn limitation does reduce silicic acid: nitrate utilization in comparison to conditions of Fe limitation, more widespread Mn limitation in a glacial Southern Ocean would be consistent with the elevated residual silicic acid: nitrate concentration ratios predicted by the geochemical proxies.

## Methods

**Fieldwork and sample analyses**. Field sampling and experiments were conducted onboard the *RRS James Clark Ross* in November 2018 (JR18002) and followed previously published protocols[57]. Seawater was collected under trace-metal-clean conditions using a towed water sampling device (~2 m depth) and filled into 1 L acid-washed polycarbonate bottles (Nalgene). Triplicate amendments of Fe, Mn, and Fe+Mn (2 nM for each amendment) were performed and were incubated for 2–5 days (Supplementary Table 1), with generally longer durations for experiments in colder waters allowing for more time for slower growth responses to become observable following supply of limiting nutrient(s). Fe and Mn spikes were prepared in 0.01 M HCl (Fisher Optima Grade HCl diluted in Milli-Q water) using 99 + % purity salts ($FeCl_3$ and $MnCl_2$). In addition, three bottles were incubated with no amendment (controls) and three were sampled for initial conditions.

Approximate mean irradiances were calculated for in situ and experimental conditions. Mean mixed layer irradiances ($\bar{E}_{ML}$) were calculated using the following equation:

$$\bar{E}_{ML} = \frac{E_0}{K_d}\left(1 - e^{K_d MLD}\right), \qquad (1)$$

Where $E_0$ is the average incident photosynthetically active radiation irradiance over the incubation duration, $K_d$ is the diffuse downwelling attenuation coefficient that was derived from experimental site chlorophyll-a concentrations using a Southern Ocean specific equation[58], and MLD is the mixed layer depth measured at conductivity-temperature-depth locations adjacent to experimental seawater sampling sites. Mixed layer depths were calculated as the depth at which density

increased by 0.01 kg m$^{-3}$ relative to a reference density at 2 m depth. Mean experimental irradiances were calculated by attenuating $E_0$ using expected transmission fractions through the incubator screening (0.35; Lee Filters "Blue Lagoon"), the polymethyl methacrylate (Perspex) incubator (0.92), and the polycarbonate incubator bottles (0.85). The ratio between these estimates of in situ and experimental irradiances are shown in Supplementary Table 1 for each experiment.

Chlorophyll-a concentrations were determined on glass fiber filters that were extracted in 90% acetone using a calibrated Turner Designs Trilogy fluorometer. Flow cytometry samples (2 mL) were preserved with paraformaldehyde (1% final concentration) and stored at –80 °C prior to analysis in the home laboratory on a FACSCalibur flow cytometer with CellQuest version 3.3 software (Becton Dickenson, Oxford, United Kingdom) following methods described in ref. [57]. A FASTOcean Fast Repetition Rate fluorometer (Chelsea Technologies Group) was used to determine $F_v/F_m$. Fluorescence transients were fit in FASTPro8 software (Chelsea Technologies Group) to yield minimum fluorescence ($F_o$) and maximum fluorescence ($F_m$). Blank fluorescence of 0.2 μm sample filtrates was subtracted from $F_o$ and $F_m$ before calculation of $F_v/F_m = (F_m–F_o)/F_m$. Nutrient samples for all experiments apart from Experiment 8 were analyzed on ship using a nutrient autoanalyzer (AA3, Seal Analytical). Certified Reference Material (CRM; Kanso Technos, Japan) was used to check for accuracy and calculate detection limits and precision. Analysis of reference material Kanso CD yielded concentrations of (mean ± SD, $n = 28$) nitrate = 5.56 ± 0.08 μM; silicic acid = 14.53 ± 0.08 μM, and phosphate = 0.47 ± 0.004 μM. Detection limits calculated as 3 × SD of the repeat analysis of this CRM were nitrate = 0.027 μM, silicic acid = 0.05 μM, and phosphate = 0.017 μM. Precision calculated as 1RSD were nitrate = 0.54%, silicic acid = 1.47% and phosphate = 9.5%. Nutrient samples for Experiment 8 were analyzed after ~11 months frozen storage; all nutrient concentrations for this experiment showed large declines in concentrations in comparison to initial ($t = 0$ hrs) samples that were measured on ship; nutrient results for this experiment were therefore not included in Fig. 2n, o. Samples for particulate organic carbon (POC), biogenic silica (BSi), and diagnostic phytoplankton pigments were sampled via filtration of water pooled from the triplicate replicates. POC samples were collected on pre-combusted glass fiber filters, subject to concentrated hydrochloric acid fuming to remove particulate inorganic carbon, and analyzed on a Eurovector EA3000 Elemental Analyzer with Callidus version 5.1 software. The blank for POC analyses was determined via taking unused pre-combusted glass fiber filters through the entire procedure, yielding 0.40 ± 0.55 μmol carbon (mean ± SD, $n = 8$). The detection limit for the POC analyses was determined as 3 × SD of these procedural blanks (1.66 μmol carbon). BSi samples were collected on 0.8 μm polycarbonate filters (Whatman), digested in 0.2 M NaOH for 2 h at 90 °C, neutralized with 0.1 M HCl, and analyzed using a nutrient autoanalyzer (QuAAtro, Seal Analytical). The blank for the BSi analyses was determined via taking unused polycarbonate filters through the entire procedure, yielding 0.061 ± 0.043 μM (mean ± SD, $n = 8$). The detection limit for the BSi analyses determined as 3 × SD of these procedural blanks was 0.62 μM. Pigment samples were collected on glass fiber filters, extracted in 90% acetone, and analyzed using high performance liquid chromatography (Dionex UltiMate 3000 LC system with Chromeleon version 7.0 software, Thermo Scientific)[59]. Pigment standards were acquired from Sigma-Aldrich (USA) and the International Agency for 14 C Determination (Denmark). Pigments concentrations were converted to approximate phytoplankton types using CHEMTAX[60] using Group 3 starting pigment ratios in ref. [61].

Dissolved trace element samples were collected at the beginning of experiments via in-line filtration through a 0.45/0.2 μm filter unit (Sartorius Sartobran 300), acidified with 140 μL Optima grade HCl (Fisher Scientific), and analyzed following the method of ref. [62] except that a Preplab (PS Analytical) was used for automated sample pre-concentration and standard addition was used to quantify both Mn and Fe. A GEOTRACES intercalibration run included in the analytical run (GSP91) produced concentrations of 0.14 ± 0.02 nM Fe and 0.69 ± 0.08 nM Mn (mean ± SD, $n = 3$), which agrees with consensus values (https://www.geotraces.org/standards-and-reference-materials/) and other reported measurements of this material[63,64]. The detection limits, determined as 3 × SD of a blank (a low concentration seawater sample diluted 1:100 in acidified Milli-Q water, $n = 6$), were 0.038 nM for DFe and 0.015 nM for DMn. The precision determined via the replicate analysis of the previously described GSP91 intercalibration standard were 12.8% for DFe and 12.3% for DMn (1RSD, $n = 3$).

**Modeling**. Two simple models were used to reinforce our suggested controls on DMn and DFe, which lead to observed limitation patterns. The first simulates changes in DFe and DMn in sub-photic zone waters initiated with elevated Fe–Mn dissolved concentration pairs from the data set of refs. [15,47] located around the Antarctic Peninsula. Changes in DFe are described by:

$$\frac{dDFe}{dt} = DFe_{remin} - DFe_{scav}, \qquad (2)$$

Where $DFe_{remin}$ is DFe derived from remineralization of sinking particles and $DFe_{scav}$ is DFe removed by scavenging processes. Remineralized DFe is determined from changes in sinking particulate organic matter with depth, where a fixed sinking C flux at 100 m depth was attenuated to a specified depth using the Martin formulation[65]. Conversion from remineralized C to Fe used assumed-average

elemental stoichiometry of particulate organic matter[2]. Derivation of scavenged Fe followed a common approach used in biogeochemical models, via first partitioning the DFe pool between free DFe ($Fe\prime$) and organically complexed DFe ($FeL$) using a prescribed ligand concentration (set to 0.8 nM) and a conditional stability coefficient (set to $10^{11}$ M$^{-1}$; e.g., ref. [66]):

$$DFe = Fe\prime + FeL \tag{3}$$

$$\beta = FeL/Fe\prime \cdot L\prime, \tag{4}$$

where $\beta$ is the conditional stability coefficient and $L\prime$ is the free ligand concentration. Only the $Fe\prime$ pool is subject to scavenging, which is determined by a prescribed rate constant ($k_{scav} = 0.001$ d$^{-1}$ after refs. [67,68]):

$$DFe_{scav} = k_{sc}Fe\prime. \tag{5}$$

Changes in DMn were calculated by:

$$\frac{dDMn}{dt} = DMn_{remin} - DMn_{redox}, \tag{6}$$

where $DMn_{remin}$ is DMn derived from remineralization of sinking particulate organic matter and $DMn_{redox}$ is DMn removed or added by redox processes. Remineralization was prescribed as for DFe. Redox processes were resolved using the equations of ref. [4], by simulating changes in solid Mn oxides (MnOx; see Eq. 8) alongside DMn, and subjecting both to exchange via:

$$DMn_{redox} = -k_{ox}DMn + k_{red}MnOx, \tag{7}$$

where $k_{ox}$ is the rate constant for oxidation of DMn to MnOx and $k_{red}$ is the rate constant for reduction of MnOx to DMn. Changes in MnOx were calculated by:

$$\frac{dMnOx}{dt} = MnOx_{sink} - MnOx_{redox}, \tag{8}$$

where $MnOx_{redox}$ was calculated following:

$$MnOx_{redox} = k_{ox}DMn - k_{red}MnOx. \tag{9}$$

Sinking Mn oxides, $MnOx_{sink}$, was calculated following the approach of ref. [4]. For MnOx concentrations below 25 pM, the sinking velocity is constant at 1 m d$^{-1}$. For MnOx concentrations >25 pM, the sinking velocity of MnOx increases linearly from 1 m d$^{-1}$ at the ocean surface (set here to 1 m depth) and 10 m d$^{-1}$ at 5500 m depth (note that ref. [4] used a 2500 m depth threshold and assumed a constant 10 m d$^{-1}$ sinking rate for all greater depths). Note that for the simulations in Fig. 3b, c, with high starting DMn concentrations, MnOx remains above this threshold. For the simulations in Fig. 3b, c, $k_{ox}$ and $k_{red}$ were set to the same values as ref. [16], with $k_{ox} = 0.0082$ d$^{-1}$ and $k_{red} = 0.041$ d$^{-1}$ (refs. [3,4,7]).

Simulations for Fig. 3b, c were conducted by initiating the model with a suite of DFe-DMn concentration pairs measured on the same samples from refs. [15,47], located around the Antarctic Peninsula. In order that these concentrations best-approximated waters in recent contact with shelf sediments, we only used concentration pairs from these datasets at sites with <750 m bathymetry and located deeper than 200 m in the water column. The bathymetric boundary (750 m) corresponds approximately to the shelf edge zone where surface radium concentrations measured in the same region transition from high shelf values to low off-shelf values (indicating strong dilution with open ocean waters[48]). The water column depth criteria of >200 m depth was used so that concentration pairs were less impacted by biological processes in surface waters. Figure 3b, c shows the suite of simulation results for each DFe-DMn concentration pair in the dataset fulfilling these criteria. We chose to show a simulation time of 20 years, which covers the time period that waters would be advected off shelf (days–weeks) as well as visualizing longer term tendencies. The simulation depth was set to those of the trace element observations (variable). For simplicity, we set the remineralization flux to 0 in the displayed simulations (true value unknown; Fig. 3b, c); setting this to higher values always led to further increases the time at which Mn* stays elevated (see below with regards to sensitivity analyses). The initial MnOx concentration was set to $0.2 \times$ DMn nM, reflecting the approximate DMn: MnOx stoichiometry that has been measured in deep waters[3,4].

The simulations show that shelf waters stay Fe deficient with respect to Mn (i.e., positive Mn*) for periods much longer than the expected timescales for off-shelf transport (days–weeks[48,49]). Modifying Mn oxidation rates in the simulations by 50% shifted the timescale at which Mn* approaches 0 from several years (default oxidation rate doubled) to >20 years (default oxidation rate halved) time range, either of which is much longer than the expected off-shelf transport (days–weeks[48,49]). Likewise, increasing maximum Mn oxide sinking rates to much higher values than expected (from 10 m d$^{-1}$ to 40 m d$^{-1}$; cf. measurements by ref. [69] indicating values <10 m d$^{-1}$ and mostly <1 m d$^{-1}$) shortens the time period before Mn* approaches 0, but this still remains several years or more in most cases. Because of the elevated starting Mn concentrations in these waters, this conclusion is also not particularly sensitive to the choice of the critical Mn oxide concentration aggregation threshold (whereupon settling velocities of Mn oxides through the water column cease to increase with depth; default from ref. [4] of MnOx = 25 pM). Reducing this threshold by a factor of 10 leads to no changes in the simulation, as Mn oxide concentrations remain much higher than this. Increasing the threshold by a factor of 10 leads to very low particulate Mn sinking, and consequently Mn*

shows no reduction over the simulation timescale. Altering the default Fe ligand characteristics (changing the concentration to 0.6 or 1 nM or conditional stability coefficient to $10^{12}$ M$^{-1}$) also led to minor changes in Mn*, which as for the default simulations remained positive for extended periods in either case. Likewise, increasing the remineralization flux from the default of 0 (for example, to 4.6 mmol C m$^{-2}$ d$^{-1}$ measured by ref. [70] over the Antarctic Peninsula shelf using $^{234}$Th; different methods used in ref. [70] ranged = 0.26–8.2 mmol C m$^{-2}$ d$^{-1}$) progressively increases the timescale at which Mn* stays positive (i.e., even more time for off-shelf transport).

Once deep waters are transported to the surface, biological activity and altered redox processes have the potential to strongly impact DFe and DMn dynamics. To investigate this, we simulated the biogeochemical response to surface entrainment of (i) the mean shelf-associated DFe and DMn concentrations for the Antarctic Peninsula dataset described previously[15,47] (DFe = 2.39 nM; DMn = 1.78 nM), and (ii) the mean of deep (1000–2000 m) water column GEOTRACES DFe and DMn measurements made in central Drake Passage and further east in a remote sector of the Atlantic Southern Ocean[16,40,43,44] (DFe = 0.42 nM; DMn = 0.17 nM; locations shown as crosses in Fig. 3a panel; depth range 1000–2000 m). We modified the two-layer "slab" nutrient-phytoplankton-zooplankton-detritus ecosystem model of ref. [50] to include DFe, DMn, and MnOx as a platform for our experiments. Fe scavenging and Mn redox processes were parameterized as for the first model, except the Mn oxide reduction rate constant, $k_{red}$, was modified to account for higher values under incident sunlight (see below). Biological uptake and recycling of DFe and DMn was prescribed as for the default nutrient in the model (nitrate) using assumed-average micronutrient quotas of phytoplankton to covert between them (0.129 moles N: $6.05 \times 10^{-5}$ moles Fe: $2.26 \times 10^{-5}$ moles Mn)[2]. Phytoplankton specific growth rate in the model is controlled by temperature, the availability of light, and nutrient availability (the model nutrient limitation term was modified to a minimizing Michaelis–Menten form for nitrate, DFe, or DMn, which subsequently sets the nutrient limitation term $L_N$; Fig. 3d, f). Phytoplankton in the model are subject to grazing and mortality, of which a portion of biomass sinks or enters the detrital pool and then the dissolved nutrient pool (i.e., representing nutrient recycling). The parameterization for all these processes were set to the defaults of ref. [50]. The model is forced by seasonality in mixed layer depth, incident irradiance, and temperature for the location of interest; in this study, mixed layers were taken from a climatology[71] for central Drake Passage, sea surface temperatures from World Ocean Atlas, and incident irradiances were calculated for 58°S. Mixed layer deepening resupplies surface waters with nutrients from the deep 'slab' (concentrations fixed at those previously described), which are then taken up by phytoplankton growth in spring. Results presented in Fig. 3d–g are for a seasonal cycle after 3 years simulation (whereupon a steady repeating seasonal cycle had been reached). To assess the potential impact of modified Mn oxide reduction rates, simulations were run for (i) the dark reduction rate (0.040 d$^{-1}$)[3,4,7], (ii) the high light reduction rate (2.35 d$^{-1}$)[7], and (iii) a reduction rate scaled linearly between (i) and (ii) as set by the mean mixed layer light level in the model (see Fig. 3d–g where shading shows induced variability).

To assess the model response to increased DFe and DMn input via dust, reflecting expected interglacial-glacial changes, we repeated ecosystem model simulations with an additional dust input term (Supplementary Fig. 6). For these simulations, we fixed the Mn oxide reduction rate to mean mixed layer light values. To reflect the interglacial dust flux to the Southern Ocean we used the low modern-day estimate of 0.014 g m$^{-2}$ yr$^{-1}$ (ref. [9]) and for the glacial dust flux we used 3.45 g m$^{-2}$ yr$^{-1}$ (ref. [24]), which match modeled ranges[72]. The total Fe and Mn content of dust were set to average continental crust values (Fe = 30,890 p.p.m.; Mn = 527 p. p.m.)[27], which is similar to the bulk Fe:Mn ratio of southern South Atlantic aerosols[28] and Patagonian dust[29]. Solubilities of Fe and Mn measured for dust vary widely: we used the median (Fe = 6.8%; Mn = 20%) and interquartile range (Fe = 3.1–13.5%; Mn = 6.5–39.2%) from the compilation of southern South Atlantic and Southern Ocean aerosol solubility measurements of ref. [28]. The model was ran using the extremes of these interquartile ranges (i.e., lower quartile for Fe and upper quartile for Mn and vice versa) to generate an uncertainty range (shown by shading in Supplementary Fig. 6). Modifying dust deposition fluxes between the chosen glacial-interglacial values (which represent expected end members) simulated values in between the two scenarios. Sensitivity runs where the dust was applied in discrete pulses (monthly or quarterly rather than the default of even deposition over a year) led to more abrupt concentration changes but changes in Mn deficiency remained.

**Reporting summary**. Further information on research design is available in the Nature Research Reporting Summary linked to this article.

## Data availability
Experimental data are provided in Supplementary Table 1. Additional Southern Ocean trace metal data sets used in this study are available in public repositories: the GEOTRACES Intermediate Data Product 2017 via the British Oceanographic Data Centre (https://www.bodc.ac.uk/geotraces/data/idp2017/) and the Antarctic Peninsula dataset via the Biological and Chemical Oceanography Data Management Office (https://www.bco-dmo.org/dataset/3800/data; https://www.bco-dmo.org/dataset/3801/data). Source Data are provided with this paper.

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

## Acknowledgements

We thank the captain, crew, and principal scientist (Yvonne Firing) of the *RRS James Clark Ross* JR18002 cruise. R. Tuerena, M. Humphreys, and A. Annett are thanked for assistance at sea. We are grateful to T. Steffens, D. Jasinski, S. Khoo, A. Mutzberg, T. Klüver, and K. Nachtigall for technical assistance and I. Rapp for advice on the trace element measurements. A. Brearley and M. Meredith are thanked for facilitating cruise participation.

## Author contributions

T.J.B. designed the study, performed the data analysis, and wrote the manuscript. T.J.B. and E.M. performed the fieldwork. E.P.A. oversaw the trace element analyses. A.E. oversaw the flow cytometry analyses. All authors commented on the manuscript.

## Funding

## Competing interests

The authors declare no competing interests.
