## [Peer Review File · Nature Communications]

REVIEWER COMMENTS

Reviewer #1 (Remarks to the Author):

Manganese co-limitation of phytoplankton growth and major nutrient drawdown in the Southern Ocean
Browning et al.

This manuscript presents results from ten bioassays developed in the Southern Ocean (Drake Passage) demonstrating Mn limitation / co limitation (Fe-Mn) in this region. The experiments are well developed, including a nice suite of measurements (e.g. Chl a, differences in POC, BSi accumulation, Fv/Fm, etc.) and replicate analysis. Results presented by the authors agree and expand the knowledge about Mn limitation in the Southern Ocean, doing a good job integrating bioassay insights with available data from the GEOTRACES program, unveiling potentially Mn / Mn-Fe stressed areas. The first part of the manuscript is well written and organized, linking the new data with previous work done in the area and with the spatial distribution of Mn* tracer. However, the transition from the bioassay experiments to the broad statements regarding the productivity cycling in the Southern Ocean and the posited reconciliation of macronutrient utilization in the geological past does not flow smoothly, reads out of context based on the provided introduction, and the conclusions drawn are rather weak and seem speculative with the data presented (simplified model).

Specific comment and suggestions are listed below identified by line numbers.

Lines 8-10: I would not express it as an inefficiency... Not a proper term to refer to natural process. Variability of PP levels/ rates etc., not efficiency related.

Line 19: Conversely to what?

Lines 28-29: What do you mean by absolute? Please rephrase.

Lines 38-40: I am aware of length restriction for the journal; however, more introductory information is needed to follow what you are describing in this manuscript (Mn cycling in the Southern Ocean and glacial / inter-glacial primary productivity shift and its relationships with dust delivery). As it is, the intro does not set the basis for what is being discussed in the manuscript.

Lines 50-58: Figure 1, in the experiments a, b and c (Fe limited), although no significant differences have been observed between the addition of Fe vs the addition of Fe+Mn, why Fe+Mn tend to yield lower Chl-a levels than just Fe addition? It will be worth to expand on this. For experiments 5 and 7 (Mn limited based on the manuscript), why just Mn limited if Fe also increases significantly primary productivity? For those Fe limited experiments, additions of Mn did not increase Chl-a significantly as Fe did for those experiments labeled as Mn limited. Please clarify this and clearly state what authors means by "limiting micronutrients".

Line 60: How do the biochemical and physiological parameters change when Fe is added for experiments 5 and 7 (Mn-limited)? Do they increase? To a similar extent followed by the addition of Mn... Probably it would be illustrative to have a similar plot in the extended section displaying the changes of the biochemical and physiological parameters following the supply of BOTH Fe and Mn. This will provide the actual differences between Fe and Mn additions.

Line 61: Fig. 1 not 11.

Lines 62-63: Connected to the previous comment, what is driving the enhanced macronutrient drawdown after Mn addition?

Line 64: Fig. 1n, o, the labels for the lower panels plots were wrong. Please correct them.

Lines 73-76: Please rephrase this statement for clarity.

Line 76: How do phytoplankton behaves when Fe⁺ and Mn are added together? Do Fv/Fm and BSi accumulation increase?

Lines 77-80: Likewise, more explanation is needed here to guide the readers as well as comparison with results from Fe and Mn experiments, not just how silicic acid drawdown and BSi accumulation evolve followed Mn addition at the Mn limited sites relative to Fe supply at Fe limited sites.

Lines 94-96: Not sure how predictable is this, while Extended Data Figure 4 display a great number of calculated Mn* (observations from other publications and GEOTRACES intermediate data product), the places where those observations match actual bioassays/incubation experiments are restricted. Where bioassays found Fe limitation there are no mixed layer Mn* values. I would be more careful with the statement "were predictable on the basis of measured dissolved Fe and Mn".

Lines 99-100: High Mn* values were also found in station 10, which was neither Fe nor Mn limited; this should be stated in the text and discussed.

Line 134: remove/ replace the word exaggerated...increased extended??

Lines 134-136: Mn won't be scavenged? Parekh et al. do not mention a differential scavenging rate of these two elements. What about the presence of organic ligands enhancing Fe solubility?

Lines 151-156: This paragraph seems rather out of place here, while the manuscript is describing natural cycling/processes of Mn and Fe (linked to primary production), why bring human-mediated fertilization experiments without further explanation?

Lines 168-173: I do not think these conclusions can be drawn based on the new data presented in the manuscript (bioassay experiments) and the simplified model employed here. Or they will need further explanation.

Line 174: Regarding the panels k-r in Figure 1, the labels are wrong, as well the reference of each panel in the manuscript (as noted in previous comments). Also, are the +Fe and +Mn X-axis labels located in the extremes of the x-axis by any reason? I would put them in the middle to ease interpretations, or alternatively remove x-labels and ad a legend stating red symbols +Fe, blue symbols +Mn, etc.

Would it be possible to add the number of the experiments besides each line -where lines are superimposed numbers can be added together-.

Line 193: replace p by q to be consistent with the figure, or re-label the figure.

Line 210: Based on what you have trebled the concentrations?

Line 583: Can you please add y axis ticks to all the plots to ease reading.

Line 616: Published solubility of Mn are widely dispersed (based on the sources, processing and characteristics of the receiving water) from ~10 to more than 90 % (Baker et al. 2006b; Buck et al. 2013; Chester et al. 1993; Hsu et al. 2005; Shelley et al. 2012; Mahowald et al., 2018 among many others). To make this more informative it should encompass a wide range of solubility, and/or provide the basis on why did you select this solubility value.

Line 330: "Fieldwork and sample analyses" section: Where are the QA/QC for other analysis besides dFe and dMn? What is the precision of the biochemical and physiological analyses (POC, Si, Nutrients, etc.)? All these is needed.

Also, blanks, detection limits, etc. for the analyses are not shown in the manuscript, especially for trace metal ones.

Line 335: I do not know if I missed it, but where is stated the amount of Fe and Mn added in the experiments, and the rationale behind this addition.

Reviewer #2 (Remarks to the Author):

Review for: "Manganese co-limitation of phytoplankton growth and major nutrient drawdown in the Southern Ocean"

In this manuscript, Browning and colleagues present evidence for limitation of Southern Ocean phytoplankton by the metal manganese (Mn). This evidence comes from factorial bio-assay experiments where iron and Mn are added individually and in combination. From a transect in the Southern Ocean, they show Mn limitation and Mn-Fe co-limitation in open ocean waters that likely have not interacted with margin sediments from Antarctica or South America. Meanwhile, Fe-limitation predominates at locations closer to these continents, due to slower Mn removal kinetics.

I think this will prove to be a very important manuscript. Although Mn limitation was more or less ignored during the past decades of iron fertilization experiments, there has been increased speculation about Mn limitation recently, but no definitive evidence. This definitive evidence has now been supplied by Browning et al., and will likely spur a re-evaluation of Southern Ocean biogeochemistry, carbon cycling, and paleoceanography.

I have a few small comments regarding the manuscript in its present state.

1) Given the spatial pattern of incubations, it's hard to justify why the map of experiments is not shown in the main text. This will help readers navigate the nuances in experimental results much more easily.

2) The discussion and presentation of the model results (Lines 113–149 and figure 2) is quite compressed and could either be simplified considerably, or elaborated upon to explain fully. The main goal of the section seems to be in the demonstration that waters with recent shelf interaction will lead to Fe limitation while waters without shelf interaction will lead to Mn limitation.

With this in mind, the discussion of dFe stability and dMn stability in Atlantic deep waters is unnecessary and somewhat misleading. The authors argue that the scavenging kinetics of Fe and Mn pre-dispose deep ocean waters to be Mn deficient. Thus, when they upwell in the SO, Mn limitation manifests. However, the model of van Hulst et al. (2016: ref 16), from which Mn oxidation rates are derived are explicitly tuned to deepwater concentrations by shutting off scavenging below a threshold value of 0.125 nM. So over long timescales, Mn* is not determined by the rate of scavenging so much as the de facto imposition of an inert Mn-binding ligand in this model, whose existence has not been documented in deep waters at these concentrations. A similar argument could be made for the role of the simulated Fe ligand concentration in setting deep ocean Mn*.

The point here though is that these nuances are totally unnecessary to the authors' goal and their ecosystem model could just be initialized by the mean characteristics of these deep water masses. I'd recommend this section (Fig. 2A and the text describing it) be rewritten or removed from the main text. This mostly applies to the 'longer term' simulations. These problems are less impactful for the

short term scenario in Fig 2.A.c and 2.A.d because concentrations are higher overall and less controlled by model ligand concentration.

Line comments.

97: It would be helpful to discuss actual concentrations of Fe and Mn in relation to previous measurements before transforming them into derived quantities like Mn^* . For instance, both the dFe and dMn concentrations here seem higher than 'typical' Southern Ocean surface values in the Geotraces IDP2017 database (e.g. Fig S4) so discussion that scales Mn^* thresholds to these datasets would be informative.

332: given the role of light in SO productivity and Fe/Mn requirements, some comments on experimental and environmental light levels would be good to add. Here or to table 1.

335: some explanation for this variation in experiment length would be welcome.

367-369: New community consensus values for GEOTRACES standards are available and should be probably be invoked for validation (<https://www.geotraces.org/standards-and-reference-materials/>).

Fig 1. Legend: Dashed line is in q not p

Figure 2: The lettering scheme for figure 2 doesn't really make sense as both A and B have sub panels a-d. It would much more legible to have a-h.

Extended Table 1. It seems misleading to use ICP-MS counts to indicate uncertainty in metal concentrations because these are not independent measurements. These should be presented without uncertainties if only extracted once.

Extended data figures 4, 5. It can be a bit hard to see the transition to negative Mn^* in this figure. It seems like this was chosen to have a symmetrical color axis but this could be made more legible if modified. The perception here is that Mn is as depleted as Fe, not more so, as argued in text.

Reviewer #3 (Remarks to the Author):

his paper provides evidence that Mn can be limiting to phytoplankton growth in the Southern Ocean. A series of 10 experiments across the Drake Passage showed Mn limitation in two sites, and co-limitation with Fe in one other. The remaining site showed mostly iron limitation. The sites with Mn (co)limitation were in the middle of the passage, while those closer to S America or to the Antarctica Peninsula were Fe limited. They also find changes in phytoplankton photophysiology consistent with different limiting conditions. I am not an expert of micronutrients, but these do seem like important results.

The authors' hypothesis that the different source of the waters to these sites, and different timescales of stripping Mn or Fe from the system are consistent with these results. This hypothesis seems reasonable given the evidence of deep water having Mn deficiencies and assuming that waters near a land mass might be more influenced by terrestrial sources. I was less convinced by the modelling aspect of the paper. Though the model was only briefly mentioned in the main text, a substantial part of the methods section was devoted to the model. I found this a bit at odds – possibly the manuscript I am reading is a revision from a version that had much more about the model in the main text? But it seems odd and not quite right to have so much buried in the methods. And because of this the last two paragraphs came across as more speculative and not well backed up.

My major concerns are:

- What is the source water to the euphotic zone along this transect? The model component uses results from an Atlantic transect as suggested source waters. But given the position of the transect downstream of the Drake passage, I would expect a much stronger imprint of Southern Ocean waters, and in particular the Pacific sector. If it is demonstrably North Atlantic water that is coming into the euphotic zone in the Mn limited stations, then this should be provided as proof in the paper. I think there needs to be some discussion about where the source water are coming from.
- Model results: long term simulation - I think 2a (and b) do rather nicely show that the timescales do lead to Mn deficient waters along the Atlantic transect. And as such is a good exploration of the age of water being important to being Mn limited. I found the setup of this experiment informative about time scaling for the loss of the different microelement. However, given my point above, is this relevant for the stations examined?
- Model results: short term simulation. I found this simulation much less convincing. Why increase all nutrient 3 times? Would results change if only twice, or if different factors were used for the different nutrients? What is the Fe:Mn ratios of "terrestrial inputs"/"shelf sediments"? The model was run 3 years – this seemed arbitrary. If the experiment was only 2 months long, then Fe would not have been as limiting relative to Mn. What are the real timescales for water to interact with the continental shelves and make it into the euphotic layer? Without providing some convincing timescales to this, I could really accept these results as convincing for the particular stations under investigation. Though indeed the results are interesting from a more academic view point that in explaining the experimental results.
- Model: ecosystem model: I guess the results are not very surprising given the source waters that are provided – but see my first point above.

Specifics:

Title: I wasn't quite sure why the title include "and major nutrient drawdown in the Southern Ocean", this doesn't seem like an emphasis of the paper?

Line 23-24: The lack of Mn limitation is not the only reason why model predictions might not be accurate – what about other micro-nutrients (e.g. B12, Cu, Zn etc). i.e. I find this sentence a bit bold.

Line 98: what is the assumed R_{Fe:Mn}? Moore et al 2013 (referenced here for the value) appears to only provide a range of each relative to C. Since Fe quotas for phytoplankton range widely (2 orders of magnitude according the Moore et al 2013) and Mn at least 1 order of magnitude – is it really realistic to be able to use such a ratio in this way? Might be worth a sentence or so to convince the reader.

Line 107-108: The figure the authors refer to here is a transect of the North Atlantic. Is the upwelled water in this transect purely North Atlantic water? Given the ACC direction I would believe that the source waters in this region would have strong Pacific SO sector imprint. What are Southern Ocean values of Mn*?

Line 146-149: this is opaque without a careful read of the methods.

Line 156-158: As with comment for line 23-24: But models are also wrong in that they don't consider a whole host of other micronutrients, some of which have been shown to be (co) limiting in other regions. Thus though this sentence is not wrong, it seems to make more of the importance of Mn than necessary. Though if the authors disagree, then a better case should be made the Mn is the next most important microelement to consider in model development.

Line 159 onwards: this seems too much speculation for which to end the paper. The "proof" is to refer to a modelling figure in the extended data – this seems a bit thin to warrant. Either bring the model more into the main text or leave out completely. These experiments (i.e. ED Fig 6) are not even described in the methods (though some information is given in the caption)

Extended material:

Line 447: It is unlikely that the c flux along the whole transect remains the same – how sensitive are results if you had more reasonable increases/decreases as the water went through subtropical gyres, upwelling zones etc?

Lines 459-462: As mentioned above, I find the initial conditions arbitrary. How sensitive are the results to this choice of 3 times increase?

Line 488-489: Could you have used nutrient/micronutrient initial conditions of the deep water from the actual stations themselves in the model?

Line 498-500: This is a bit opaque and could be elaborated.

Lines 610-620: How sensitive is the model to the many choices of dust concentrations, solubilities etc of these experiments.

ED Figure 1: this is a nice figure – I would suggest including the main body of the paper.

ED Fig 2: maybe state that this was initial conditions?

Response to Reviewer comments

We thank the three Reviewers for their detailed comments. We have responded to these below in blue.

To refer between review responses, we have numbered reviewer comments (R1_1, for Reviewer 1, comment 1, and so on).

Reviewer #1 (Remarks to the Author):

Manganese co-limitation of phytoplankton growth and major nutrient drawdown in the Southern Ocean

Browning et al.

This manuscript presents results from ten bioassays developed in the Southern Ocean (Drake Passage) demonstrating Mn limitation / co limitation (Fe-Mn) in this region. The experiments are well developed, including a nice suite of measurements (e.g. Chl a, differences in POC, BSi accumulation, Fv/Fm, etc.) and replicate analysis. Results presented by the authors agree and expand the knowledge about Mn limitation in the Southern Ocean, doing a good job integrating bioassay insights with available data from the GEOTRACES program, unveiling potentially Mn / Mn-Fe stressed areas. The first part of the manuscript is well written and organized, linking the new data with previous work done in the area and with the spatial distribution of Mn⁺ tracer. However, the transition from the bioassay experiments to the broad statements regarding the productivity cycling in the Southern Ocean and the posited reconciliation of macronutrient utilization in the geological past does not flow smoothly, reads out of context based on the provided introduction, and the conclusions drawn are rather weak and seem speculative with the data presented (simplified model).

We thank the Reviewer for their comments. We have implemented a number of changes in the revised manuscript in response to their recommendations, including adding further details to the introduction and revising the last section of the manuscript.

Specific comment and suggestions are listed below identified by line numbers.

R1_1:

Lines 8-10: I would not express it as an inefficiency... Not a proper term to refer to natural process. Variability of PP levels/ rates etc., not efficiency related.

We have now rephrased this to: *“Residual macronutrients in the surface Southern Ocean result from restricted biological utilization”*

R1_2:

Line 19: Conversely to what?

We have now removed the word ‘conversely’. Revised text is: *“We found manganese (co-)limited phytoplankton growth and macronutrient consumption in central Drake Passage, whilst iron limitation was widespread nearer the South American and Antarctic continental shelves.”*

R1_3:

Lines 28-29: What do you mean by absolute? Please rephrase.

We have now replaced 'absolute' with 'essential'. Revised text is "*All of Earth's photosynthetic organisms have an essential Mn requirement for the water splitting reaction in photosynthesis*"

R1_4:

Lines 38-40: I am aware of length restriction for the journal; however, more introductory information is needed to follow what you are describing in this manuscript (Mn cycling in the Southern Ocean and glacial / inter-glacial primary productivity shift and its relationships with dust delivery). As it is, the intro does not set the basis for what is being discussed in the manuscript.

We have now expanded the introduction, adding two extra paragraphs to provide additional detail about Mn sources and cycling in the Southern Ocean and the potential role of dust in driving glacial-interglacial Southern Ocean primary productivity changes (lines 31–58 in the revised manuscript).

R1_5:

Lines 50-58: Figure 1, in the experiments a, b and c (Fe limited), although no significant differences have been observed between the addition of Fe vs the addition of Fe+Mn, why Fe+Mn tend to yield lower Chl-a levels than just Fe addition? It will be worth to expand on this.

This is an interesting point—although the mean chlorophyll-a values are statistically indistinguishable from one another (there is overlap in the concentration ranges of the triplicate biological replicates between +Fe and +Fe+Mn), the mean values are indeed lower for +Fe+Mn addition at these sites. We speculate that the experimental increase of Mn by 2 nM under Fe limited conditions could potentially increase competition for Fe uptake at cell membrane transporter sites, thereby possibly decreasing the amount of Fe uptake (e.g. Sunda and Huntsman, 1998). However, to our knowledge competition between these two elements specifically has not been observed for phytoplankton, plus the impact of this might be expected to be more acute in the +Mn treatment, which showed similar chlorophyll-a values to the controls (a decrease would be expected if Fe uptake capacity was markedly reduced). Although we agree that this is potentially an interesting avenue for exploration, as the noted differences are not significant (statistically), we decided not to provide discussion/interpretation on this in the revised manuscript.

Reference:

Sunda, W.G. and Huntsman, S.A. Processes regulating cellular metal accumulation and physiological effects: phytoplankton as model systems. *Science of the Total Environment* **219**, 165-181 (1998).

R1_6:

For experiments 5 and 7 (Mn limited based on the manuscript), why just Mn limited if Fe also increases significantly primary productivity? For those Fe limited experiments, additions of Mn did not increase Chl-a significantly as Fe did for those experiments labeled as Mn limited. Please clarify this and clearly state what authors means by "limiting micronutrients".

This is indeed correct. We suggest here that this was caused by either independent limitation of two parts of the community (limited by Mn and Fe, respectively) or a degree of

substitutability in Fe and Mn demand in superoxide dismutase enzymes (Saito et al., 2008; Peers and Price, 2004).

We have now revised the section of text where we first define the observed limitation categories to further discuss this point:

“For the experiments classified as primary Mn limited (Fig. 2e, g), small but significant increases in chlorophyll-a within +Fe treatments were also observed. This implied some degree of Mn-Fe co-limitation at these sites, which could have been caused by either, or a combination of, (i) stimulation of different communities, each limited by one of the nutrients (either Mn or Fe), and/or (ii) substitutability of Fe and Mn in superoxide dismutase enzymes³¹, meaning that supply of either nutrient would reduce requirement for the other^{32,37}.”

References:

- Peers, G. & Price, N.M. A role for manganese in superoxide dismutases and growth of iron-deficient diatoms. *Limnol. Oceanogr.* **49**, 1774-1783 (2004).
- Saito, M.A., Goepfert, T.J. and Ritt, J.T. Some thoughts on the concept of colimitation: three definitions and the importance of bioavailability. *Limnol. Oceanogr.* **53**, 276-290 (2008).

R1_7:

Line 60: How do the biochemical and physiological parameters change when Fe is added for experiments 5 and 7 (Mn-limited)? Do they increase? To a similar extent followed by the addition of Mn... Probably it would be illustrative to have a similar plot in the extended section displaying the changes of the biochemical and physiological parameters following the supply of BOTH Fe and Mn. This will provide the actual differences between Fe and Mn additions.

The summary figure of biological and physiological responses (original Figure 1k–r; revised Figure 2k–q) indicates changes between +Fe and +Mn treatments, such that a positive slope indicates a greater increase in the parameter for +Mn over +Fe. However, as the Reviewer points out, it does not indicate how +Fe changes relative to control values. In the following we briefly outline details of responses and describe a new figure we have added to the Supplementary Information (new Supplementary Fig. 1). In Experiment 5, changes in phosphate and silicic acid concentrations following Fe addition remained statistically indistinguishable from controls (whilst in +Mn they showed significant drawdown; ANOVA $p < 0.05$, followed by Fisher Least Significant Difference test). F_v/F_m was statistically indistinguishable from control values for both +Fe and +Mn treatments (but increased following +Fe+Mn). POC concentrations declined a small amount in the +Fe treatment, whilst BSi increased (whilst both were larger in the +Mn treatment). Fucoxanthin concentrations in the +Fe treatment remained similar to control values (but increased in the +Mn treatment). In Experiment 7, phosphate and silicic acid concentration were drawn down from control values following Fe addition (although in +Mn they showed a significantly larger drawdown). Mean F_v/F_m increased relative to the control in the +Fe treatment, but this remained statistically indistinguishable from the +Mn treatment. POC and fucoxanthin concentrations increased in the +Fe treatment (but not as much as for +Mn), whilst BSi showed little change (with the +Mn treatment showing an increase).

This data is displayed in a new Supplementary Figure (Supplementary Data Fig. 1; this figure is restricted to the Mn limited Experiments 5 and 7 for clarity/simplicity). This figure, alongside Figure 2k–q (original Fig. 1k–r), is now referenced in the main text whenever discussing these parameters.

Supplementary Figure 1. Changes in biochemical and physiological parameters at Mn limited Experiments 5 and 7. a–g, Experiment 5; h–n, Experiment 7. Where available in triplicate biological replicates, bar heights indicate the mean, white symbols are the individual replicate values, and different letters above bars indicate statistically different mean responses (ANOVA $p < 0.05$, followed by Fisher Least Significant Difference test). Bars with no points or letters have only one replicate (pooled treatments). Arrows indicate average initial values ($n=3$ for chlorophyll-a and F_v/F_m , $n=1$ for POC and fucoxanthin concentrations).

R1_8:

Line 61: Fig. 1 not 11.

We were actually intending to refer to Figure 1, subplot (letter) 'i'. This is now Figure 2 in the revised manuscript (i.e., Fig. 2i).

R1_9:

Lines 62-63: Connected to the previous comment, what is driving the enhanced macronutrient drawdown after Mn addition?

We interpret the nutrient drawdown as a result of phytoplankton growth following supply of the limiting micronutrient (e.g., Mn).

This has been clarified in the revised manuscript: *“Dissolved macronutrient drawdown during incubations matched biomass increases, with Mn additions at the two Mn limited sites producing significantly enhanced macronutrient drawdown as a result of biomass formation relative to either untreated controls or +Fe amendment (Fig. 2n, o; Supplementary Fig. 1).”*

R1_10:

Line 64: Fig. 1n, o, the labels for the lower panels plots were wrong. Please correct them.

These have been corrected.

R1_11:

Lines 73-76: Please rephrase this statement for clarity.

This has now been simplified to: *“More restricted increases in F_v/F_m following recovery from Mn limitation in +Mn treatments likely reflected phytoplankton community transitions from Mn limitation into Fe limitation following Mn supply.”*

R1_12:

Line 76: How do phytoplankton behaves when Fe+ and Mn are added together? Do F_v/F_m and BSi accumulation increase?

Please refer to our response to #R1_7.

R1_13:

Lines 77-80: Likewise, more explanation is needed here to guide the readers as well as comparison with results from Fe and Mn experiments, not just how silicic acid drawdown and BSi accumulation evolve followed Mn addition at the Mn limited sites relative to Fe supply at Fe limited sites.

We have amended this section to make our point clearer. We have also added an additional Supplementary Figure (Supplementary Fig. 2) showing silicic acid: nitrate concentration ratios for all experiments and treatments, which we refer to (new figure also reproduced below).

The revised text reads:

“The latter was potentially linked with changes in silicic acid: nitrate concentration ratios in the experiments (Supplementary Fig. 2). As previously observed for Fe amendments under Fe limited conditions^{38,39}, silicic acid: nitrate ratios increased over controls following Fe addition in Experiments 2, 4, and 9 (changes in Experiments 1 and 3 were insignificant), resulting from reduced silicic acid utilized per unit phytoplankton biomass produced. Conversely, in Mn limited Experiments 5 and 7, silicic acid: nitrate ratios decreased following Mn addition relative to controls and +Fe treatments, whilst calculated diatom contributions showed little or no differences (contributions to chlorophyll-a of 78–79% (Fe–Mn) and 84% (Fe and Mn) in Experiments 5 and 7, respectively). Whilst this observation in itself could be readily explained by Mn supply at these sites rapidly switching the main limiting nutrient from Mn to Fe, and thereby driving Fe-stress-induced increases silicic acid utilized per biomass formation^{38,39}, the +Fe+Mn treatments also showed reduced silicic acid: nitrate ratios. The latter observation suggesting that a transition to either Fe limited or (micro)nutrient replete conditions led to increased silicic acid: nitrate drawdown.

Supplementary Figure 2. Changes in dissolved silicic acid: nitrate concentration ratios (Si:N) in experiments. Bar heights indicate the mean, white symbols are the individual replicate values, and different letters above bars indicate statistically different mean responses (ANOVA $p < 0.05$, followed by Fisher Least Significant Difference test; n.s. indicates not significant). Results for Experiment 8 are not shown as samples were stored frozen for >1 year prior to analysis (see Methods; they however showed significant silicic acid: nitrate drawdown in +Fe and +Fe+Mn additions relative to controls and +Mn).

R1_14:

Lines 94-96: Not sure how predictable is this, while Extended Data Figure 4 display a great number of calculated Mn^* (observations from other publications and GEOTRACES intermediate data product), the places where those observations match actual bioassays/incubation experiments are restricted. Where bioassays found Fe limitation there are no mixed layer Mn^* values. I would be more careful with the statement “were predictable on the basis of measured dissolved Fe and Mn”.

For our specific set of cross-Drake Passage experiments, responses to Fe or Mn supply were predictable from dissolved Fe and Mn in the form of calculated Mn^* (i.e., the Mn limited sites had the lowest Mn^* , -0.02 to 0.04 nM, whilst Mn^* at the Fe limited sites ranged 0.16 – 0.31 nM; Supplementary Table 1). Indeed, as the Reviewer points out we cannot make assessments of predictability in other regions of the Southern Ocean, as there are insufficient bioassays with associated Mn^* . However, if we assume the link we found between Mn^* and Fe versus Mn limitation in Drake Passage are representative of the wider Southern Ocean and extrapolate using other measurements of near-surface Mn^* , it suggests broad regions open ocean regions where Mn^* is low and therefore likely limiting or co-limiting alongside Fe (revised Supplementary Fig. 5). A caveat in relating such normalized excess concentrations, pointed out by the Reviewer in the next comment (R1_15), is that it is possible for concentrations of both Fe and Mn to be high and therefore neither of them limiting (e.g., light limited instead); in this case Mn^* predicts the nutrient that would likely become limiting following biological drawdown of both nutrients. We have made minor revisions to the paragraph to add this detail (please refer to next comment #R1_15).

R1_15:

Lines 99-100: High Mn^* values were also found in station 10, which was neither Fe nor Mn limited; this should be stated in the text and discussed.

As mentioned in the response to R1_14, the Reviewer points out it is possible for concentrations of both Fe and Mn to be high and therefore neither of them limiting (e.g., light limited instead); in this case Mn* predicts the nutrient that would likely become limiting following biological drawdown of both nutrients. This detail has been added to the revised manuscript:

“Mn was also elevated at the one site located in close proximity to Elephant Island (2.35 nM), where concentrations of both DFe and DMn were high (>2 nM) and phytoplankton were not limited by either nutrient (Fig. 2j). Under the latter conditions, Mn* relates to which of either Fe or Mn would be expected to become limiting following continued biological drawdown⁴².”*

R1_16:

Line 134: remove/ replace the word exaggerated...increased extended??

We have changed ‘exaggerated’ to ‘extended’.

R1_17:

Lines 134-136: Mn won't be scavenged? Parekh et al. do not mention a differential scavenging rate of these two elements. What about the presence of organic ligands enhancing Fe solubility?

Our reference to Parekh et al. was intended to relate specifically to the enhanced scavenging of Fe that they predict with higher particulate organic matter concentrations (i.e., more particle surface for Fe to be scavenged to). Indeed, Mn is not believed to be scavenged in the same way to particle surfaces. As the Reviewer points out, it might well be the case that in surface waters there is more ligand binding of Fe and this might act to reduce scavenging and potentially balance out the effect of increased particle surface area for Fe scavenging. As the relative importance of these two controls has not been clearly demonstrated we have removed this part so that the revised manuscript text simply states that differences in removal rate in surface waters will be extended by Mn photoreduction.

R1_18:

Lines 151-156: This paragraph seems rather out of place here, while the manuscript is describing natural cycling/processes of Mn and Fe (linked to primary production), why bring human-mediated fertilization experiments without further explanation?

An important implication of finding Mn(-Fe) (co-)limitation is that additions of Fe alone to Southern Ocean seawater will not maximize phytoplankton growth in such cases. So we feel that it is relevant to point this out briefly in this section, using purposeful anthropogenic Fe additions as one example. We have updated the wording of this text in the revised manuscript to:

“Our finding of Mn limitation is important for understanding the drivers of productivity and associated feedbacks in the Southern Ocean^{11,25}. Most immediately, Mn (co-)limitation throughout the extensive Mn deficient regions of the Southern Ocean (Supplementary Fig. 5) would restrict the fertilization impact of any mechanism exclusively enhancing the supply of Fe to these regions. This could include mechanisms such as anthropogenic Fe supply⁴⁵ or increased Fe ligand concentrations⁵¹. Furthermore, as current ocean-climate models do not include Mn this would also be an important factor, among others^{52,53}, in restricting the accuracy of their predicted responses to such forcing in these regions.”

R1_19:

Lines 168-173: I do not think these conclusions can be drawn based on the new data presented in the manuscript (bioassay experiments) and the simplified model employed here. Or they will need further explanation.

We have now updated this section to more completely describe the potential impact of (i) enhanced dust deposition in the glacial Southern Ocean on Mn limitation, and (ii) the potential relevance of altered silicic acid: nitrate ratios that were observed following Fe or Mn additions (at sites limited by each nutrient) to geochemical proxy records of these nutrients. In this revised section we describe more explicitly how and with which caveats we arrive at these conclusions. Certainly, with the data to hand these ideas remain hypotheses, but we feel this is an appropriate place in the manuscript to put these ideas forward. The paragraphs added to the revised manuscript are:

“It is possible that Southern Ocean Mn limitation was more prevalent in the past. In glacial periods, enhanced dust Fe is thought to have increased diatom growth⁵⁴ and hence nitrate utilization⁵⁵, potentially driving a major fraction of the atmospheric CO₂ drawdown recorded in ice cores^{24–26}. Typical continental-derived dust Fe:Mn ratios exceed assumed average phytoplankton requirements >20-fold^{2,27}, implying that, with other factors staying the same, greater dust deposition might strengthen Mn limitation, in addition to stimulating overall increases in productivity and major nutrient drawdown. Reported solubilities of Mn and Fe in dust are wide-ranging: a recent compilation of southern South Atlantic and Southern Ocean solubilities found interquartile ranges of 3.1–13.5% for Fe (n=19) and 6.5–20% for Mn (n=25)²⁸. Calculations with the extremes of these solubility ranges, applied to typical continental crust concentrations²⁷, indicate a Mn deficient dust source in each case (range of 1.7–44.7 fold higher soluble Fe:Mn than assumed-average phytoplankton requirements). In the ocean, the impact of this dust flux on Mn would be modulated by mixing with deep waters. If we repeat our simple ecosystem model simulation for the open ocean scenario (deep water Mn*=0.015 nM; Fig. 3e), but include elevated glacial dust deposition rates with the observed solubility ranges, we find that Mn limitation intensifies (Mn* becomes negative) under all scenarios (Supplementary Fig. 6).*

A more Mn limited glacial Southern Ocean could potentially be linked to the observed changes in residual silicic acid: nitrate ratios in our experiments. We found that Fe supply at Fe limited sites generally enhanced remaining concentration ratios, matching earlier observations in Fe-limited systems^{38,39}, whereas ratios decreased following Mn supply at the Mn limited sites (Supplementary Fig. 2). This indicated that conditions of either Mn or Fe limitation are potentially an important control on the relative drawdown of these two nutrients. If this is indeed the case, this is pertinent because whilst geochemical proxies suggest enhanced Southern Ocean nitrate utilization⁵⁵, they also indicate silicic acid was not drawdown⁵⁶, which is at odds with increased diatom growth⁵⁴. If conditions of Mn rather than Fe limitation do reduce silicic acid utilization relative to nitrate, an enhanced extent and intensity of Mn limitation in a more productive glacial Southern Ocean would be consistent with the elevated residual silicic acid: nitrate concentration ratios predicted by the geochemical proxies.”

R1_20:

Line 174: Regarding the panels k-r in Figure 1, the labels are wrong, as well the reference of each panel in the manuscript (as noted in previous comments). Also, are the +Fe and +Mn X-axis labels located in the extremes of the x-axis by any reason? I would put them in the middle to ease interpretations, or alternatively remove x-labels and add a legend stating red symbols +Fe, blue symbols +Mn, etc.

The letter labels have been corrected in the revised manuscript. The +Fe and +Mn label locations have a function, in that the slopes of the lines in these subplots summarise differences between +Fe addition (always set to 0) and +Mn. Red lines and symbols represent the Fe limited experiments and blue the Mn limited experiments, as indicated by the chlorophyll-a responses in panels a–j.

R1_21:

Would it be possible to add the number of the experiments besides each line -where lines are superimposed numbers can be added together-

Experiment numbers added as suggested.

R1_22:

Line 193: replace p by q to be consistent with the figure, or re-label the figure.

Corrected in the revised manuscript.

R1_23:

Line 210: Based on what you have trebled the concentrations?

This section of the modelling has been revised. We refer the Reviewer to Reviewer 2 comment #R2_3 and our subsequent response for full details. In brief, the short timescale simulations have been now been initiated with Fe-Mn concentration pairs from a set of observations from the Antarctic Peninsula shelf region (Hatta et al., 2013; Measures et al., 2013). In order that concentrations best approximated waters in recent contact with shelf sediments, we only used concentration pairs from these datasets at sites with less than 750 m bathymetry and located deeper than 200 m in the water column. The bathymetric boundary (750 m) corresponds approximately to the shelf edge zone where surface radium concentrations measured in the same region transition from high shelf values to low off shelf values (indicating strong dilution with open ocean waters; Dulaiova et al., 2009). The water column depth criteria of >200 m depth was used so that concentration pairs were less impacted by biological processes in surface waters. In our revised figure, we chose to show the suite of simulation results for each DFe-DMn concentration pair in the dataset fulfilling these criteria. We increased the simulation timescale from 3 to 20 years, in order to better visualize the timescales over which manganese deficiency (Mn^*) in such waters tend towards zero. The longer-term simulation has been removed from the revised manuscript due to uncertainties in parameterizations that control DMn concentrations at very low concentrations.

R1_24:

Line 583: Can you please add y axis ticks to all the plots to ease reading.

Tick marks added.

R1_25:

Line 616: Published solubility of Mn are widely dispersed (based on the sources, processing and characteristics of the receiving water) from ~10 to more than 90 % (Baker et al. 2006b; Buck et al. 2013; Chester et al. 1993; Hsu et al. 2005; Shelley et al. 2012; Mahowald et al., 2018 among many others). To make this more informative it should encompass a wide range of solubility, and/or provide the basis on why did you select this solubility value.

We have now revised solubilities in these simulations to the median and interquartile ranges reported for a compilation of Fe and Mn solubilities determined on aerosols from the southern South Atlantic and Southern Ocean (Chance et al., 2015). The median solubilities of this compilation are Fe=6.8%; Mn=20% and interquartile ranges are Fe=3.1–13.5% and Mn=6.5–20%. The impact of applying these ranges are discussed in the revised manuscript text (see response to comment #R1_19 and revised Supplementary Fig. 6). Our overall finding that enhanced dust fluxes in glacial can lead to more Mn deficient, i.e., lower Mn*, conditions is robust to this solubility range, although it does indeed introduce significant ranges in simulated values.

Reference

Chance, R., Jickells, T.D. and Baker, A.R. Atmospheric trace metal concentrations, solubility and deposition fluxes in remote marine air over the south-east Atlantic. *Mar. Chem.*, **177**, 45-56 (2015).

R1_26:

Line 330: “Fieldwork and sample analyses” section: Where are the QA/QC for other analysis besides dFe and dMn? What is the precision of the biochemical and physiological analyses (POC, Si, Nutrients, etc.)? All these is needed.

Also, blanks, detection limits, etc. for the analyses are not shown in the manuscript, especially for trace metal ones.

We have extended the field/laboratory methods section and added the requested details where available.

R1_27:

Line 335: I do not know if I missed it, but where is stated the amount of Fe and Mn added in the experiments, and the rationale behind this addition.

This detail was neglected to be included in the initial manuscript. Added concentrations were 2 nM for both Fe and Mn. This has been added to the revised methods.

Reviewer #2 (Remarks to the Author):

Review for: "Manganese co-limitation of phytoplankton growth and major nutrient drawdown in the Southern Ocean"

In this manuscript, Browning and colleagues present evidence for limitation of Southern Ocean phytoplankton by the metal manganese (Mn). This evidence comes from factorial bioassay experiments where iron and Mn are added individually and in combination. From a transect in the Southern Ocean, they show Mn limitation and Mn-Fe co-limitation in open ocean waters that likely have not interacted with margin sediments from Antarctica or South America. Meanwhile, Fe-limitation predominates at locations closer to these continents, due to slower Mn removal kinetics.

I think this will prove to be a very important manuscript. Although Mn limitation was more or less ignored during the past decades of iron fertilization experiments, there has been increased speculation about Mn limitation recently, but no definitive evidence. This definitive evidence has now been supplied by Browning et al., and will likely spur a re-evaluation of Southern Ocean biogeochemistry, carbon cycling, and paleoceanography.

We thank Reviewer 2 for their positive comments and recommendations below that have led to an improved manuscript. Responses to each are provided below.

I have a few small comments regarding the manuscript in its present state.

R2_1:

1) Given the spatial pattern of incubations, it's hard to justify why the map of experiments is not shown in the main text. This will help readers navigate the nuances in experimental results much more easily.

The map (original Extended Data Figure 1) has now been included as Figure 1 in the revised manuscript.

R2_3:

2) The discussion and presentation of the model results (Lines 113–149 and figure 2) is quite compressed and could either be simplified considerably, or elaborated upon to explain fully. The main goal of the section seems to be in the demonstration that waters with recent shelf interaction will lead to Fe limitation while waters without shelf interaction will lead to Mn limitation.

With this in mind, the discussion of dFe stability and dMn stability in Atlantic deep waters is unnecessary and somewhat misleading. The authors argue that the scavenging kinetics of Fe and Mn pre-dispose deep ocean waters to be Mn deficient. Thus, when they upwell in the SO, Mn limitation manifests. However, the model of van Hulst et al. (2016: ref 16), from which Mn oxidation rates are derived are explicitly tuned to deepwater concentrations by shutting off scavenging below a threshold value of 0.125 nM. So over long timescales, Mn* is not determined by the rate of scavenging so much as the de facto imposition of an inert Mn-binding ligand in this model, whose existence has not been documented in deep waters at these concentrations. A similar argument could be made for the role of the simulated Fe ligand concentration in setting deep ocean Mn*.

The point here though is that these nuances are totally unnecessary to the authors' goal and their ecosystem model could just be initialized by the mean characteristics of these deep

water masses. I'd recommend this section (Fig. 2A and the text describing it) be rewritten or removed from the main text. This mostly applies to the 'longer term' simulations. These problems are less impactful for the short term scenario in Fig 2.A.c and 2.A.d because concentrations are higher overall and less controlled by model ligand concentration.

The Reviewer makes an important point that we have considered in detail. As stated by the Reviewer, in the model of van Hulst et al. (2017) DMn scavenging is effectively shut off below 0.125 nM, to yield deep water model concentrations in line with observations. This is achieved via imposing a critical Mn oxide concentration aggregation threshold, $MnO_x = 25$ pM, whereupon settling velocities of Mn oxides through the water column cease to increase with depth. This 25 pM threshold was derived by van Hulst et al. (2017) from the Mn redox rate constants in combination with a typical low DMn value of the deep ocean (0.125 nM; Section 2.2.6 in Van Hulst et al. 2017). The oxidation rate itself is derived independently from observations by Bruland et al. (1994) of DMn/PMn (PMn = particulate Mn), alongside the DMn reduction rates measured by Sunda and Huntsman (1994). As the Reviewer points out, this assumption effectively prohibits continued DMn removal below these concentrations, which, whilst conducted by van Hulst et al. (2017) to generally match deep water GEOTRACES observations, is indeed not mechanistic in its formulation. As the Reviewer also points out, this assumption of a low-level background concentration has little impact on Mn dynamics in these simulations at higher DMn concentrations.

To address this concern, we followed the suggestion of the Reviewer and modified this part of the modelling section. The focus in the revised manuscript in terms of simulations is on the shorter-term dynamics of Mn versus Fe removal at higher concentrations, for instance after contact with sedimentary trace element sources, in addition to the ecosystem model simulations. As the Reviewer points out, this is the most central aspect to our argument (that waters interacting with the shelf regions will retain high positive Mn^* for extended periods). In the revised manuscript we have now modified these shorter duration simulations in several ways: firstly, rather than choosing some generally representative elevated Fe and Mn concentrations, we initialized the model with a suite of Fe-Mn concentration pairs measured on the same samples from the detailed studies of Hatta et al. (2013) and Measures et al. (2013) located around the Antarctic Peninsula (locations shown as circles in the new Fig. 3a map). In order that concentrations best approximated waters in recent contact with shelf sediments, we only used concentration pairs from these datasets at sites with less than 750 m bathymetry and located deeper than 200 m in the water column. The bathymetric boundary (750 m) corresponds approximately to the shelf edge zone where surface radium concentrations measured in the same region transition from high shelf values to low off shelf values (indicating strong dilution with open ocean waters; Dulaiova et al., 2009). The water column depth criteria of >200 m depth was used so that concentration pairs were less impacted by biological processes in surface waters. In our revised figure for this section (Fig. 3b,c in the revised manuscript), we chose to show the suite of simulation results for each DFe-DMn concentration pair in the dataset fulfilling these criteria. A second change implemented is that we increased the simulation timescale from 3 to 20 years, in order to better visualize the timescales over which manganese deficiency (Mn^*) in such waters tend towards 0 nM. We note in the revised manuscript text that the exact value of Mn^* reached at even longer timescales is sensitive to unresolved mechanisms that presumably act to prevent DMn descending to 0 nM. A third change we apply is that for simplicity we set the remineralization flux to 0, as values of sinking particulate organic carbon flux are quite variable and we find that setting this to values >0 further increases the time at which Mn^* stays elevated (noted below with regards to sensitivity analyses).

The results of this reworked analysis show that in these deeper, near-shelf waters, Mn^* values are initially high in seawater (>92% DFe-DMn concentrations pairs have $Mn^* > 0.2$ nM; $n=26$) and remain so for multiple years as a result of faster DFe removal at higher concentrations (new Fig. 3b,c). This supports our interpretation in indicating that: (i) in deeper waters in proximity to shelf sediments, waters are Fe deficient relative to Mn (high Mn^*) and would therefore become Fe rather than Mn limited following biological nutrient drawdown; and (ii) it indicates that these waters would have to be isolated for extended periods before Mn would approach co-deficient levels (i.e. Mn^* around 0 nM). Although this duration is sensitive to parameters varied in our sensitivity analysis (see below), they remain much longer than expected timescales of off shelf transport (i.e., years rather than days-weeks; Dulaiova et al. 2009; Jiang et al. 2019). The latter strongly suggests that a high Mn^* signature is maintained as waters are advected off shelf.

Modifying Mn oxidation rates in the simulations by 50% shifts the timescale at which Mn^* approaches 0 from several years (default oxidation rate doubled) to >20-years (default oxidation rate halved) time range. Again, as off shelf transport is expected to be much more rapid than this (the radium study of Dulaiova et al. 2009 indicates days to weeks for off shelf transport around Elephant Island; Jiang et al. 2019 produces similar model estimates for this region), the elevated Mn^* in these waters would be expected to significantly impact off shelf waters in either case (i.e., leading to Fe rather than Mn deficient waters downstream of such sediment interaction). Likewise, increasing maximum Mn oxide sinking rates to much higher values than expected (40 m d^{-1} ; cf. measurements by Glockzin et al. 2014 indicating values $<10 \text{ m d}^{-1}$ and mostly $<1 \text{ m d}^{-1}$) shortens the time period before Mn^* approaches 0, but this still remains several years or more in most cases. Because of the elevated starting Mn concentrations in these waters, this conclusion is also not sensitive to the choice of the critical Mn oxide concentration aggregation threshold (whereupon settling velocities of Mn oxides through the water column cease to increase with depth; default from Van Hulst et al. 2017 of $MnOx=25 \text{ pM}$). Reducing this threshold by a factor of 10 leads to no changes in the simulation as particulate Mn concentrations in the simulation remain higher than this. Increasing the threshold by a factor of 10 leads to extremely low particulate Mn sinking, and consequently Mn^* shows no reduction over the simulation timescale. Altering the default Fe ligand characteristics (e.g. changing the concentration to 0.6 nM or conditional stability coefficient to 10^{12} M^{-1}) also did not change this conclusion. Likewise, increasing the remineralization flux from the default of 0 (for example, to $4.6 \text{ mmol C m}^{-2} \text{ d}^{-1}$ measured by Buesseler et al. (2010) over the Antarctic Peninsula shelf using ^{234}Th ; different methods used in Buesseler et al. (2010) ranged $=0.26\text{--}8.2 \text{ mmol C m}^{-2} \text{ d}^{-1}$) progressively increases the timescale at which Mn^* stays positive (i.e., even more time for off shelf transport). The findings from this sensitivity analysis are now noted in the model parameterization sections of the revised methods (and explicitly referred to in the main text).

In contrast to near shelf waters, deep water column GEOTRACES measurements made in central Drake Passage and further east in a remote sector of Atlantic Southern Ocean (Middag et al., 2011; 2012; Klunder et al., 2011; 2014; new Fig. 3a panel), have low Mn^* values due to extended periods away from terrestrial sources (>96% DFe-DMn concentrations pairs have $Mn^* < 0.2$ nM; $n=30$; indicated as red symbols in revised Fig. 2c). These are primed for Mn-Fe co-limitation upon transfer to the surface, provided they do not interact substantially with shelf-impacted waters previously described.

A further related modification we have implemented is that the subsequent ecosystem model simulations have been initialized with average observed deep-water DFe and DMn concentrations for the shelf and open ocean systems previously described (rather than the

end point of the long timescale simulations used previously), also as suggested by the Reviewer.

Figure 3. Controls on Fe versus Mn limitation in the Southern Ocean. **a–c**, Evolution of Mn^* in deep waters. Waters in close contact with shelf sediments (circle locations in ‘a’) have elevated Mn^* (concentrations indicated as $t=0$ values in ‘b’ and ‘c’; Methods)^{15,47}, which persists when scavenging removal is modelled for each element over the simulation timescale (lines in ‘b’ and ‘c’; Methods). In contrast, deep open ocean waters isolated from sediments (cross locations in ‘a’) have low Mn^* (red ticks in ‘c’). **d–g**, Ecosystem model simulation of additional surface ocean processes. Deep-water DFe and DMn concentrations in the model were set to: **d–e**, mean observations for shelf-impacted (circles in ‘a’), and **f–g**, open ocean waters (crosses in ‘a’). L_N is phytoplankton nutrient limitation in the model, where lower values indicates stronger nutrient limitation. Shading represents the model ranges generated when the rate constant for Mn reduction is set to either dark or high light values; solid lines are the results when the rate constant is scaled linearly between dark and light values as a function of mean mixed layer irradiance (see main text and Methods).

References:

- Dulaiova, H., Ardelan, M.V., Henderson, P.B. & Charette, M.A. Shelf-derived iron inputs drive biological productivity in the southern Drake Passage. *Global Biogeochem. Cy.* **23**, GB4014 (2009).
- Buesseler, K.O., McDonnell, A.M., Schofield, O.M., Steinberg, D.K. & Ducklow, H.W. High particle export over the continental shelf of the west Antarctic Peninsula. *Geophys. Res. Lett.* **37**, L22606 (2010).
- Glockzin, M., Pollehne, F. and Dellwig, O. Stationary sinking velocity of authigenic manganese oxides at pelagic redoxclines. *Mar. Chem.* **160**, 67-74 (2014).
- Hatta, M., Measures, C.I., Selph, K.E., Zhou, M. & Hiscock, W.T. Iron fluxes from the shelf regions near the South Shetland Islands in the Drake Passage during the austral-winter 2006. *Deep Sea Res. Pt II* **90**, 89-101 (2013).
- Jiang, M., Measures, C.I., Barbeau, K.A., Charette, M.A., Gille, S.T., Hatta, M., Kahru, M., Mitchell, B.G., Garabato, A.C.N., Reiss, C. & Selph, K. Fe sources and transport from the Antarctic Peninsula shelf to the southern Scotia Sea. *Deep Sea Res. Pt I* **150**, 103060 (2019).

- Klunder, M.B., Laan, P., Middag, R., De Baar, H.J.W. & Van Ooijen, J.C. Dissolved iron in the Southern Ocean (Atlantic sector). *Deep Sea Res. Pt II* **58**, 2678-2694 (2011).
- Klunder, M.B. et al. Dissolved Fe across the Weddell Sea and Drake Passage: impact of DFe on nutrient uptake. *Biogeosciences* **11**, 651-669 (2014).
- Measures, C.I., Brown, M.T., Selph, K.E., Apprill, A., Zhou, M., Hatta, M. & Hiscock, W.T. The influence of shelf processes in delivering dissolved iron to the HNLC waters of the Drake Passage, Antarctica. *Deep Sea Res. Pt II* **90**, 77-88 (2013).
- Middag, R.D., De Baar, H.J.W., Laan, P., Cai, P.V. & Van Ooijen, J.C. Dissolved manganese in the Atlantic sector of the Southern Ocean. *Deep Sea Res. Pt II* **58**, 2661-2677 (2011).
- Middag, R., De Baar, H.J.W., Laan, P. & Huhn, O. The effects of continental margins and water mass circulation on the distribution of dissolved aluminum and manganese in Drake Passage. *J. Geophys. Res.* **117**, C01019 (2012).

Line comments.

R2_4:

97: It would be helpful to discuss actual concentrations of Fe and Mn in relation to previous measurements before transforming them into derived quantities like Mn*. For instance, both the dFe and dMn concentrations here seem higher than 'typical' Southern Ocean surface values in the Geotraces IDP2017 database (e.g. Fig S4) so discussion that scales Mn* thresholds to these datasets would be informative.

Indeed, both DFe and DMn are a somewhat elevated with respect to other open-ocean, surface Southern Ocean concentrations. For example, we measured DFe and DMn concentrations at Experiment Sites 5–7 (central Drake Passage) between 0.34–0.44 nM for DFe and 0.14–0.17 nM for DMn, whilst GEOTRACES values from the central Drake Passage were between 0.049–0.37 nM for DFe (Klunder et al., 2014) and 0.086–0.2 nM for DMn (Middag et al., 2012). We attribute this to our observations being at the beginning of the growth season in spring (November) whilst the GEOTRACES observations were made in later summer (April), thereby allowing for several months of biological drawdown in combination with reduced frequency of deep mixing events. The same applies to trace element measurements from other studies in Drake Passage (Martin et al., 1990; Browning et al., 2014) and elsewhere in open ocean waters (Klunder et al., 2011; Middag et al., 2011), all of which reach slightly lower DFe and DMn concentrations but which were also all conducted in mid-later summer (February–April). Our early occupation also resulted in enhanced chlorophyll-a concentrations in Drake Passage with respect to late summer (e.g. Thomalla et al., 2011; Browning et al., 2014), suggesting phytoplankton demand tracks decreasing DFe and DMn availability through the season. We have now added a comparison of our trace element values to these other datasets in an additional paragraph in the revised manuscript:

“The springtime dissolved Fe (DFe) and dissolved Mn (DMn) concentrations in central Drake Passage were slightly elevated with respect to previous observations, which were all made in mid-late summer^{13,16,18,40} (Supplementary Table 1; cf. central Drake Passage GEOTRACES observations of DFe=0.049–0.37 nM⁴⁰ and DMn=0.086–0.2 nM¹⁶). Chlorophyll-a concentrations were also elevated with respect to typical later season values^{18,41}. We attribute these differences to continued biological drawdown of both Fe and Mn through the growth season, with biological demand (approximated by phytoplankton biomass) tracking availability of these nutrients.”

References

- Browning, T.J. et al. Strong responses of Southern Ocean phytoplankton communities to volcanic ash. *Geophys. Res. Lett.* **41**, 2851-2857 (2014).
- Klunder, M.B., Laan, P., Middag, R., De Baar, H.J.W. & Van Ooijen, J.C. Dissolved iron in the Southern Ocean (Atlantic sector). *Deep Sea Res. Pt II* **58**, 2678-2694 (2011).
- Klunder, M.B. et al. Dissolved Fe across the Weddell Sea and Drake Passage: impact of DFe on nutrient uptake. *Biogeosciences* **11**, 651-669 (2014).
- Martin, J.H., Gordon, R.M. & Fitzwater, S.E. Iron in Antarctic waters. *Nature* **345**, 156-158 (1990).
- Middag, R.D., De Baar, H.J.W., Laan, P., Cai, P.V. & Van Ooijen, J.C. Dissolved manganese in the Atlantic sector of the Southern Ocean. *Deep Sea Res. Pt II* **58**, 2661-2677 (2011).
- Middag, R., De Baar, H.J.W., Laan, P. & Huhn, O. The effects of continental margins and water mass circulation on the distribution of dissolved aluminum and manganese in Drake Passage. *J. Geophys. Res.* **117**, C01019 (2012).
- Thomalla, S.J., Fauchereau, N., Swart, S. & Monteiro, P.M.S. Regional scale characteristics of the seasonal cycle of chlorophyll in the Southern Ocean. *Biogeosciences* **8**, 2849–2866 (2011).

R2_5:

332: given the role of light in SO productivity and Fe/Mn requirements, some comments on experimental and environmental light levels would be good to add. Here or to table 1.

We calculated the mean average photosynthetically available radiation (PAR) over the experiment incubation duration for (i) estimated mixed layers at CTD locations adjacent to experimental seawater sampling sites, and (ii) in the on-deck incubators. These calculations have been added to the revised methods (reproduced below). We report the difference between these estimates of in situ and experimental irradiances in Table S1 as $E_{in\ situ}/E_{exp}$ for each experiment. These fractions range from 0.46 (i.e. irradiance in situ estimated as 46% of experimental irradiance) to 1.99, mainly driven by variability in mixed layer depths between the different sites.

New irradiance section in revised Methods:

“Approximate mean irradiances were calculated for in situ and experimental conditions. Mean mixed layer irradiances (\bar{E}_{ML}) were calculated using the following equation:

$$\bar{E}_{ML} = \frac{E_0}{K_d} (1 - e^{-K_d MLD}), \quad (1)$$

Where E_0 is the average incident PAR irradiance over the incubation duration, K_d is the diffuse downwelling attenuation coefficient that was derived from experimental site chlorophyll-a concentrations using a Southern Ocean specific equation⁵⁸, and MLD is the mixed layer depth measured at conductivity-temperature-depth (CTD) locations adjacent to experimental seawater sampling sites. Mixed layer depths were calculated as the depth at which density increased by 0.01 kg m⁻³ relative to a reference density at 2 m depth. Mean experimental irradiances were calculated by attenuating E_0 using expected transmission fractions through the incubator screening (0.35; Lee Filters “Blue Lagoon”), the polymethyl methacrylate (Perspex) incubator (0.92), and the polycarbonate incubator bottles (0.85). The ratio between these estimates of in situ and experimental irradiances are shown in Supplementary Table 1 for each experiment.”

Reference

Venables, H. & Moore, C.M. Phytoplankton and light limitation in the Southern Ocean: Learning from high-nutrient, high-chlorophyll areas. *J. Geophys. Res. Oceans* **115**, C02015 (2010).

R2_6:

335: some explanation for this variation in experiment length would be welcome.

Experiments were run for longer in colder waters, with the expectation that growth responses to supply of limiting nutrient(s) would take longer to become observable. This has been added to the revised methods.

“Triplicate amendments of Fe, Mn, and Fe+Mn (2 nanomoles for each amendment) were performed and were incubated for 2–5 days (Supplementary Table 1), with generally longer durations for experiments in colder waters allowing for more time for slower growth responses to become observable following supply of limiting nutrient(s).”

R2_7:

367-369: New community consensus values for GEOTRACES standards are available and should be probably be invoked for validation (<https://www.geotraces.org/standards-and-reference-materials/>).

We have added this comparison to the revised methods section (our GSP values match consensus values for Fe and Mn in addition to the other cited studies).

R2_8:

Fig 1. Legend: Dashed line is in q not p

This has been corrected.

R2_9:

Figure 2: The lettering scheme for figure 2 doesn't really make sense as both A and B have sub panels a-d. It would much more legible to have a-h.

We have changed this to use a continuous letter sequence.

R2_10:

Extended Table 1. It seems misleading to use ICP-MS counts to indicate uncertainty in metal concentrations because these are not independent measurements. These should be presented without uncertainties if only extracted once.

We agree and have now removed these.

R2_11:

Extended data figures 4, 5. It can be a bit hard to see the transition to negative Mn* in this figure. It seems like this was chosen to have a symmetrical color axis but this could be made more legible if modified. The perception here is that Mn is as depleted as Fe, not more so, as argued in text.

We have now changed the colour scheme in revised Extended Data Figure 4 (now Supplementary Fig. 5) so that Mn* values around 0 nM are yellow and can be more clearly identified. The original Extended Data Figure 5 has been removed from the revised manuscript.

Supplementary Figure 5. Wider extent of Mn deficiency in the surface Southern Ocean. Values are calculated with <30 m depth data from the GEOTRACES Intermediate Data Product², CLIVAR^{3,4}, and Refs 5–11. Yellow points indicate sites approaching Mn-Fe co-deficiency. Red crosses indicate locations where experiments (replicated bottle-scale bioassay experiments or mesoscale enrichment experiments) have found evidence for Fe limitation (from compilation of Ref. 1).

Reviewer #3 (Remarks to the Author):

This paper provides evidence that Mn can be limiting to phytoplankton growth in the Southern Ocean. A series of 10 experiments across the Drake Passage showed Mn limitation in two sites, and co-limitation with Fe in one other. The remaining site showed mostly iron limitation. The sites with Mn (co)limitation were in the middle of the passage, while those closer to S America or to the Antarctica Peninsula were Fe limited. They also find changes in phytoplankton photophysiology consistent with different limiting conditions. I am not an expert of micronutrients, but these do seem like important results.

The authors' hypothesis that the different source of the waters to these sites, and different timescales of stripping Mn or Fe from the system are consistent with these results. This hypothesis seems reasonable given the evidence of deep water having Mn deficiencies and assuming that waters near a land mass might be more influenced by terrestrial sources. I was less convinced by the modelling aspect of the paper. Though the model was only briefly mentioned in the main text, a substantial part of the methods section was devoted to the model. I found this a bit at odds – possibly the manuscript I am reading is a revision from a version that had much more about the model in the main text? But it seems odd and not quite right to have so much buried in the methods. And because of this the last two paragraphs came across as more speculative and not well backed up.

We thank Reviewer 3 for their useful comments that have improved the manuscript. We provide responses to each of their comments and recommendations below. This includes description of a reworked modelling section that more clearly aligns with its intended purpose, namely, context for interpreting our experimental results and making predictions and hypotheses at larger spatial-temporal scales. We further integrate more discussion of model simulation details, caveats, and implications into the main manuscript text.

My major concerns are:

R3_1:

- What is the source water to the euphotic zone along this transect? The model component uses results from an Atlantic transect as suggested source waters. But given the position of the transect downstream of the Drake passage, I would expect a much stronger imprint of Southern Ocean waters, and in particular the Pacific sector. If it is demonstrably North Atlantic water that is coming into the euphotic zone in the Mn limited stations, then this should be provided as proof in the paper. I think there needs to be some discussion about where the source water are coming from.

Source waters to the mixed layer in Drake Passage represents a mixture of upwelled water from the deep Atlantic, Pacific, and Indian Oceans; exact contributions are difficult to estimate (Tamsitt et al., 2017). An important fraction of upwelling likely originates in Drake Passage itself (Viglione and Thompson, 2016), but some will indeed be upwelled in Pacific and Indian Ocean sectors and then be transported east. We note this in the revised manuscript (lines 171–173). We agree that the original manuscript would have benefited from more discussion, as suggested by the Reviewer. However, in the revised manuscript we have now removed the long timescale simulation and focus on the shorter-intermediate timescale evolution of DFe and DMn concentrations. This was due to uncertainties in model parameterization which impact the values of DMn at very low DMn concentrations. Such mechanism(s) are much less important at higher DMn concentrations characterizing shelf waters. We refer Reviewer 3 to Reviewer 2 comment #R2_3 and our response for an in-depth discussion. In the revised manuscript the new shorter timescale simulations have been (i) initialized with observed deep water DFe and DMn concentration pairs around the

Antarctic Peninsula, therefore representing waters in recent contact with continental shelf micronutrient sources, and (ii) extended to 20 years. We still discuss in the revised text how the trajectory of Mn* in the shorter timescale simulation will play out over longer timescales, in conjunction with observations of deep water DMn and DFe in the open Southern Ocean that represent the end point of this process. Further discussion of the simulations is also provided in response to the later comment #R3_3.

References

- Tamsitt, V., Drake, H.F., Morrison, A.K., Talley, L.D., Dufour, C.O., Gray, A.R., Griffies, S.M., Mazloff, M.R., Sarmiento, J.L., Wang, J. & Weijer, W. Spiraling pathways of global deep waters to the surface of the Southern Ocean. *Nat. Commun.* **8**, 172 (2017).
- Viglione, G.A. & Thompson, A.F., 2016. Lagrangian pathways of upwelling in the Southern Ocean. *J. Geophys. Res. Oceans* **121**, 6295-6309 (2016).

R3_2:

- Model results: long term simulation - I think 2a (and b) do rather nicely show that the timescales do lead to Mn deficient waters along the Atlantic transect. And as such is a good exploration of the age of water being important to being Mn limited. I found the setup of this experiment informative about time scaling for the loss of the different microelement. However, given my point above, is this relevant for the stations examined?

As noted in the previous response (R3_1) we have now removed the 600-year time scale simulation, however we have now updated the short timescale simulation to be initiated with observed DFe and DMn concentration pairs from waters around the Antarctic Peninsula and increased our simulation timescale to 20 years. Although this is not long enough to reach the final low end-point Mn* of most of the deep ocean interior (intentional due to increasing sensitivity of DMn to the chosen parameterization at low DMn concentrations; see response to Reviewer 2 comment #R2_3 for more details), the high starting Mn* and its downward trajectory are nevertheless still clear. Together with our synthesized observations of DFe and DMn in deep open ocean waters, this supports our interpretation about how Mn versus Fe deficiency comes about in the Southern Ocean (further details provided below in response to R3_3).

R3_3:

- Model results: short term simulation. I found this simulation much less convincing. Why increase all nutrient 3 times? Would results change if only twice, or if different factors were used for the different nutrients? What is the Fe:Mn ratios of "terrestrial inputs"/"shelf sediments"? The model was run 3 years – this seemed arbitrary. If the experiment was only 2 months long, then Fe would not have been as limiting relative to Mn. What are the real timescales for water to interact with the continental shelves and make it into the euphotic layer? Without providing some convincing timescales to this, I could really accept these results as convincing for the particular stations under investigation. Though indeed the results are interesting from a more academic view point that in explaining the experimental results.

We agree with the Reviewer that these simulations required more context and reasoning. As noted in responses to R3_1 and R3_2, we have now modified this shorter duration modelling in a number of ways: firstly, rather than choosing some generally representative elevated Fe and Mn concentrations, we initialized the model with a suite of Fe and Mn concentrations measured on the same samples from the detailed studies of Hatta et al. (2013) and Measures et al. (2013) located around the Antarctic Peninsula (locations identified as symbols in revised Fig. 2a). In order that concentrations best approximated waters in recent

contact with shelf sediments, we only used concentration pairs from these datasets at sites with less than 750 m bathymetry and located deeper than 200 m in the water column. The bathymetric boundary (750 m) corresponds approximately to the shelf edge zone where surface radium concentrations measured in the same region transition from high shelf values to low off shelf values (indicating strong dilution with open ocean waters; Dulaiova et al., 2009). The water column depth criteria of >200 m depth was used so that DFe-DMn concentration pairs were less impacted by biological processes in surface waters. In our revised figure (new Fig. 3b,c), we show the suite of simulation results for each DFe-DMn concentration pair fulfilling these criteria. Secondly, we increased the simulation timescale from 3 to 20 years, in order to better visualize the timescales over which manganese deficiency (Mn^*) in such waters tends towards lower values.

The results of this reworked analysis show that in these deeper, near-shelf waters, Mn^* values are initially high in seawater (>92% DFe-DMn of the concentrations pairs in the observational dataset have $Mn^* > 0.2$ nM; n=26) and remain so for multiple years as a result of faster DFe removal at higher concentrations (new Fig. 3b,c). This supports our interpretation in indicating that: (i) in deeper waters in proximity to shelf sediments, waters are Fe deficient relative to Mn (high Mn^*) and would therefore become Fe rather than Mn limited following biological nutrient drawdown; and (ii) it indicates that these waters would have to be isolated for extended periods before Mn would approach co-deficient levels (i.e. Mn^* around 0 nM). These timescales remain much longer than expected timescales of off shelf transport (i.e., years rather than days-weeks; Dulaiova et al. 2009; Jiang et al. 2019). The latter strongly suggests that a high Mn^* signature is maintained as waters are advected off shelf. A sensitivity analysis (described in the response to #R2_3 and also in the revised Methods of the manuscript) indicates that modified model assumptions do not change this interpretation.

In contrast to near shelf waters, deep water column GEOTRACES measurements made in central Drake Passage and further east in a remote sector of Atlantic Southern Ocean (Middag et al., 2011; 2012; Klunder et al., 2011; 2014; new Fig. 3a panel), have low Mn^* values due to extended periods in the oceans interior away from terrestrial sources (>96% DFe-DMn concentrations pairs have $Mn^* < 0.2$ nM; n=30; indicated as red symbols in revised Fig. 2c). These are primed for Mn-Fe co-limitation upon transfer to the surface, provided they do not interact substantially with shelf-impacted waters previously described. A further related modification we have implemented is that the subsequent ecosystem model simulations have been initialized with average observed deep-water DFe and DMn concentrations for the shelf and open ocean systems previously described (rather than the end point of the long timescale simulations used previously).

Figure 3. Controls on Fe versus Mn limitation in the Southern Ocean. **a–c**, Evolution of Mn^* in deep waters. Waters in close contact with shelf sediments (circle locations in ‘a’) have elevated Mn^* (concentrations indicated as $t=0$ values in ‘b’ and ‘c’; Methods)^{15,47}, which persists when scavenging removal is modelled for each element over the simulation timescale (lines in ‘b’ and ‘c’; Methods). In contrast, deep open ocean waters isolated from sediments (cross locations in ‘a’) have low Mn^* (red ticks in ‘c’). **d–g**, Ecosystem model simulation of additional surface ocean processes. Deep-water DFe and DMn concentrations in the model were set to: **d–e**, mean observations for shelf-impacted (circles in ‘a’), and **f–g**, open ocean waters (crosses in ‘a’). L_N is phytoplankton nutrient limitation in the model, where lower values indicates stronger nutrient limitation. Shading represents the model ranges generated when the rate constant for Mn reduction is set to either dark or high light values; solid lines are the results when the rate constant is scaled linearly between dark and light values as a function of mean mixed layer irradiance (see main text and Methods).

References

- Dulaiova, H., Ardelan, M.V., Henderson, P.B. & Charette, M.A. Shelf-derived iron inputs drive biological productivity in the southern Drake Passage. *Global Biogeochem. Cy.* **23**, GB4014 (2009).
- Hatta, M., Measures, C.I., Selph, K.E., Zhou, M. & Hiscock, W.T. Iron fluxes from the shelf regions near the South Shetland Islands in the Drake Passage during the austral-winter 2006. *Deep Sea Res. Pt II* **90**, 89-101 (2013).
- Jiang, M., Measures, C.I., Barbeau, K.A., Charette, M.A., Gille, S.T., Hatta, M., Kahru, M., Mitchell, B.G., Garabato, A.C.N., Reiss, C. & Selph, K. Fe sources and transport from the Antarctic Peninsula shelf to the southern Scotia Sea. *Deep Sea Res. Pt I* **150**, 103060 (2019).
- Klunder, M.B., Laan, P., Middag, R., De Baar, H.J.W. & Van Ooijen, J.C. Dissolved iron in the Southern Ocean (Atlantic sector). *Deep Sea Res. Pt II* **58**, 2678-2694 (2011).
- Klunder, M.B. et al. Dissolved Fe across the Weddell Sea and Drake Passage: impact of DFe on nutrient uptake. *Biogeosciences* **11**, 651-669 (2014).
- Measures, C.I., Brown, M.T., Selph, K.E., Apprill, A., Zhou, M., Hatta, M. & Hiscock, W.T. The influence of shelf processes in delivering dissolved iron to the HNLC waters of the Drake Passage, Antarctica. *Deep Sea Res. Pt II* **90**, 77-88 (2013).

Middag, R.D., De Baar, H.J.W., Laan, P., Cai, P.V. & Van Ooijen, J.C. Dissolved manganese in the Atlantic sector of the Southern Ocean. *Deep Sea Res. Pt II* **58**, 2661-2677 (2011).

Middag, R., De Baar, H.J.W., Laan, P. & Huhn, O. The effects of continental margins and water mass circulation on the distribution of dissolved aluminum and manganese in Drake Passage. *J. Geophys. Res.* **117**, C01019 (2012).

R3_4:

- Model: ecosystem model: I guess the results are not very surprising given the source waters that are provided – but see my first point above.

We agree that the outcome of the ecosystem modelling is that the source (i.e. below mixed layer) waters strongly set the resultant phytoplankton dynamics, DFe, DMn and resultant Mn*. However, it is important to note that these simulations provide information about the impact of (i) the dynamic Mn reduction rate introduced for variable mixed layer light conditions (both buffering against DMn oxidation and dissolving additional deep water sourced particulate Mn oxides), and (ii) the biological drawdown of these elements on DFe and DMn dynamics (e.g. concentration-dependant scavenging removal) in relation to subsurface supply. In the end we find that these do not play as major a role as deep mixing, which is indeed effective at resetting mixed layer concentrations to deep values, however we feel that this is still of important value to the manuscript in testing the combined roles of these different surface ocean processes.

Specifics:

R3_5:

Title: I wasn't quite sure why the title include "and major nutrient drawdown in the Southern Ocean", this doesn't seem like an emphasis of the paper?

We feel that aspects of major/macronutrient drawdown have been brought out more in the revised manuscript and as a result have not modified it.

R3_6:

Line23-24: The lack of Mn limitation is not the only reason why model predictions might not be accurate – what about other micro-nutrients (e.g. B12, Cu, Zn etc). i.e. I find this sentence a bit bold.

We see the Reviewers point here; as they imply, models are a major simplification and many aspects of them (not just nutrients, but light availability, phytoplankton diversity, grazing dynamics, viral lysis, ocean circulation etc.) could be invoked to question the accuracy of Southern Ocean dynamics under a forced change in model Fe availability. However, the point that we were trying to bring out here is that changes in Fe availability in the Southern Ocean in current model simulations do exert major primary productivity control, because the models are essentially setting up Fe to be the limiting nutrient (e.g. Dutkiewicz et al., 2005; Oschlies et al., 2010; Resing et al., 2015; Moore et al., 2018). Prior to our findings this was justifiable on the basis that Fe was indeed the only (micro)nutrient shown to be primarily limiting in the Southern Ocean (Moore et al., 2013). However, our new results suggest that Mn availability is also of primary importance in regulating phytoplankton response to Fe supply (i.e., Fe supply can be enhanced but lead to little productivity response due to insufficient Mn). We have nevertheless tempered this sentence in the abstract and the final paragraph of the manuscript to acknowledge caveats pointed out by the Reviewer and discussed in our above response.

The revised sentence in the abstract is:

“Our results suggest an important role for manganese in modelling Southern Ocean productivity and understanding nutrient drawdown in glacial periods.”

The related revised sentence in the final section in the manuscript:

“Furthermore, as current ocean-climate models do not include Mn this would also be an important factor, among others^{52,53}, in restricting the accuracy of their predicted responses to such forcing in these regions.

References

- Dutkiewicz, S., Follows, M.J. & Parekh, P. Interactions of the iron and phosphorus cycles: A three-dimensional model study. *Global Biogeochem. Cy.* **19**, GB1012 (2005).
- Resing, J.A. et al. Basin-scale transport of hydrothermal dissolved metals across the South Pacific Ocean. *Nature* **523**, 200-203 (2015).
- Moore, C.M. et al. Processes and patterns of oceanic nutrient limitation. *Nat. Geosci.* **6**, 701-710 (2013).
- Moore, J.K., Fu, W., Primeau, F., Britten, G.L., Lindsay, K., Long, M., Doney, S.C., Mahowald, N., Hoffman, F. & Randerson, J.T. Sustained climate warming drives declining marine biological productivity. *Science* **359**, 1139-1143 (2018).
- Oschlies, A., Koeve, W., Rickels, W. and Rehdanz, K., 2010. Side effects and accounting aspects of hypothetical large-scale Southern Ocean iron fertilization. *Biogeosciences* **7**, 4017-4035 (2010).

R3_7:

Line 98: what is the assumed R_{Fe:Mn}? Moore et al 2013 (referenced here for the value) appears to only provide a range of each relative to C. Since Fe quotas for phytoplankton range widely (2 orders of magnitude according the Moore et al 2013) and Mn at least 1 order of magnitude – is it really realistic to be able to use such a ratio in this way? Might be worth a sentence or so to convince the reader.

The assumed R_{Fe:Mn} is 2.67, which is the multi-study average calculated by Moore et al. (2013). This was stated earlier in the manuscript (original line number 30), but we agree it is worth repeating here and have done so in the revised manuscript. For our purposes, calculation of Mn* using a fixed R_{Fe:Mn} is useful to indicate trends in the relative deficiency of each nutrient (as used previously, e.g., Moore, 2016; and for major nutrients e.g. Deutsch et al., 2007; Sarmiento et al., 2004). Provided it remains a constant at all sites, changing the value of R_{Fe:Mn} would change the magnitude, but not impact spatial trends, in Mn* found in our study and in our data compilation (Supplementary Fig. 5). As the Reviewer points out, variability in this value across different phytoplankton communities might be expected and this would impact the true values of excess DMn relative to DFe (Mn*). However, the proof of its success as a predictor is shown by its good match to the experimental responses to Fe and Mn supply (i.e., the Mn limited sites had the lowest Mn*, –0.02 to 0.04 nM, whilst Mn* at the Fe limited sites ranged 0.16–0.31 nM; Supplementary Table 1). This finding was our justification for using Mn* as an indicator of the tendency of different Fe-Mn concentration pairs to become either Fe or Mn(-Fe) (co-)limited following biological drawdown of both nutrients.

References

- Deutsch C., Sarmiento J.L., Sigman D.M., Gruber N., Dunne J.P. Spatial coupling of nitrogen inputs and losses in the ocean. *Nature* **445**,163–167 (2007).
- Moore, C.M. Diagnosing oceanic nutrient deficiency. *Phil. T. R. Soc. A* **374**, 20150290 (2016).

Sarmiento J.L., Gruber N., Brzezinski M.A., Dunne J.P. High-latitude controls of thermocline nutrients and low latitude biological productivity. *Nature* **427**,56–60 (2004).

R3_8:

Line 107-108: The figure the authors refer to here is a transect of the North Atlantic. Is the upwelled water in this transect purely North Atlantic water? Given the ACC direction I would believe that the source waters in this region would have strong Pacific SO sector imprint. What are Southern Ocean values of Mn*?

We refer the Reviewer to our responses to their earlier comments (#R3_1, #R3_2, #R3_3).

Line 146-149: this is opaque without a careful read of the methods.

We have now revised this sentence to add some more explanatory detail:

“Within this framework we then adjusted the rate constant for Mn reduction to either a dark value (0.040 d^{-1})^{3,4,7}, a bright light value (2.35 d^{-1})⁷, or a dynamic value scaled linearly between these two extremes using mean mixed layer light in the model.”

R3_9:

Line 156-158: As with comment for line23-24: But models are also wrong in that they don't consider a whole host of other micronutrients, some of which have been shown to be (co) limiting in other regions. Thus though this sentence is not wrong, it seems to make more of the importance of Mn than necessary. Though if the authors disagree, then a better case should be made the Mn is the next most important microelement to consider in model development.

We refer the Reviewer to our response to their earlier comment and the associated changes made to the manuscript text (#R3_6). As described there, we have rephrased the manuscript text to state that an absence of Mn in models would be an important factor, among others, in restricting the accuracy of their predicted responses to altered Fe supply.

R3_10:

Line 159 onwards: this seems too much speculation for which to end the paper. The “proof” is to refer to a modelling figure in the extended data – this seems a bit thin to warrant. Either bring the model more into the main text or leave out completely. These experiments (i.e. ED Fig 6) are not even described in the methods (though some information is given in the caption)

We have now updated this section to more completely describe the potential impact of (i) enhanced dust deposition in the glacial Southern Ocean on Mn limitation, and (ii) the potential relevance of altered silicic acid: nitrate ratios that were observed following Fe or Mn additions (at sites limited by each nutrient) to geochemical proxy records of these nutrients. In this revised section we describe more explicitly how and with which caveats we arrive at these conclusions. Certainly, with the data to hand these ideas indeed remain hypotheses, but we feel this is an appropriate place in the manuscript to put these ideas forward.

The revised manuscript text for this section is:

“It is possible that Southern Ocean Mn limitation was more prevalent in the past. In glacial periods, enhanced dust Fe is thought to have increased diatom growth⁵⁴ and hence nitrate utilization⁵⁵, potentially driving a major fraction of the atmospheric CO₂ drawdown recorded in ice cores^{24–26}. Typical continental-derived dust Fe:Mn ratios exceed assumed average

phytoplankton requirements >20-fold^{2,27}, implying that, with other factors staying the same, greater dust deposition might strengthen Mn limitation, in addition to stimulating overall increases in productivity and major nutrient drawdown. Reported solubilities of Mn and Fe in dust are wide-ranging: a recent compilation of southern South Atlantic and Southern Ocean solubilities found interquartile ranges of 3.1–13.5% for Fe (n=19) and 6.5–20% for Mn (n=25)²⁸. Calculations with the extremes of these solubility ranges, applied to typical continental crust concentrations²⁷, indicate a Mn deficient dust source in each case (range of 1.7–44.7 fold higher soluble Fe:Mn than assumed-average phytoplankton requirements). In the ocean, the impact of this dust flux on Mn* would be modulated by mixing with deep waters. If we repeat our simple ecosystem model simulation for the open ocean scenario (deep water Mn*=0.015 nM; Fig. 3e), but include elevated glacial dust deposition rates with the observed solubility ranges, we find that Mn limitation intensifies (Mn* becomes negative) under all scenarios (Supplementary Fig. 6).

A more Mn limited glacial Southern Ocean could potentially be linked to the observed changes in residual silicic acid: nitrate ratios in our experiments. We found that Fe supply at Fe limited sites generally enhanced remaining concentration ratios, matching earlier observations in Fe-limited systems^{38,39}, whereas ratios decreased following Mn supply at the Mn limited sites (Supplementary Fig. 2). This indicated that conditions of either Mn or Fe limitation are potentially an important control on the relative drawdown of these two nutrients. If this is indeed the case, this is pertinent because whilst geochemical proxies suggest enhanced Southern Ocean nitrate utilization⁵⁵, they also indicate silicic acid was not drawdown⁵⁶, which is at odds with increased diatom growth⁵⁴. If conditions of Mn rather than Fe limitation do reduce silicic acid utilization relative to nitrate, an enhanced extent and intensity of Mn limitation in a more productive glacial Southern Ocean would be consistent with the elevated residual silicic acid: nitrate concentration ratios predicted by the geochemical proxies.”

The added section to the revised methods is:

“To assess the model response to increased DFe and DMn input via dust, reflecting expected interglacial-glacial changes, we repeated ecosystem model simulations with an additional dust input term (Supplementary Fig. 6). For these simulations we fixed the Mn oxide reduction rate to median mixed layer light values. To reflect the interglacial dust flux to the Southern Ocean we used the low modern-day estimate of 0.014 g m⁻² yr⁻¹ (Ref. 9) and for the glacial dust flux we used 3.45 g m⁻² yr⁻¹ (Ref. 24), which match modelled ranges⁷². The total Fe and Mn content of dust were set to average continental crust values (Fe=30,890 p.p.m.; Mn=527 p.p.m.)²⁷, which is similar to the bulk Fe:Mn ratio of southern South Atlantic aerosols²⁸ and Patagonian dust²⁹. Solubilities of Fe and Mn measured for dust vary widely: we used the median (Fe=6.8%; Mn=20%) and interquartile range (Fe=3.1–13.5%; Mn=6.5–20%) from the compilation of southern South Atlantic and Southern Ocean aerosol solubility measurements of Ref. 28. The model was ran using the extremes of these interquartile ranges (i.e., lower quartile for Fe and upper quartile for Mn and vice versa) to generate an uncertainty range (shown by shading in Supplementary Fig. 6). Modifying dust deposition fluxes between the chosen glacial-interglacial values (which represent expected end members) simulated values in between the two scenarios. Sensitivity runs where the dust was applied in discrete pulses (monthly or quarterly rather than the default of even deposition over a year) led to more abrupt concentration changes but changes in Mn deficiency remained.”

Extended material:

R3_11:

Line 447: It is unlikely that the c flux along the whole transect remains the same – how sensitive are results if you had more reasonable increases/decreases as the water went through subtropical gyres, upwelling zones etc?

The sinking flux in the original simulation was tuned to match observed phosphate accumulation. However, as discussed in previous responses (#R3_1, #R3_2, #R3_3), we have now removed this long timescale simulation.

R3_12:

Lines 459-462: As mentioned above, I find the initial conditions arbitrary. How sensitive are the results to this choice of 3 times increase?

We agree with the Reviewer. The initial conditions for these simulations have been revised so that a suite of observed Fe-Mn concentration pairs from Antarctic Peninsula-impacted waters are now simulated. We refer the Reviewer to responses #R3_3 for further details.

R3_13:

Line 488-489: Could you have used nutrient/micronutrient initial conditions of the deep water from the actual stations themselves in the model?

Indeed this is a better approach. Our shorter-term simulations are now initialized with observations (please see response to #R3_3).

R3_14:

Line 498-500: This is a bit opaque and could be elaborated.

We have re-written this methods part to be more explicit:

“To assess the potential impact of modified Mn oxide reduction rates, simulations were ran for (i) the dark reduction rate (0.040 d^{-1})^{3,4,7}, (ii) the high light reduction rate (2.35 d^{-1})⁷, and (iii) a reduction rate scaled linearly between (i) and (ii) as set by the mean mixed layer light level in the model (see Fig. 3d–g where shading shows induced variability).”

R3_15:

Lines 610-620: How sensitive is the model to the many choices of dust concentrations, solubilities etc of these experiments.

As described in our response to R3_10, we have now conducted simulations with ranges of Southern Ocean aerosol Mn and Fe solubilities and included sensitivity tests with different modes of dust delivery (continuous versus pulsed). These are discussed in the main text and revised methods.

R3_16:

ED Figure 1: this is a nice figure – I would suggest including the main body of the paper.

The map (original Extended Data Fig. 1) has now been included as Figure 1 in the revised manuscript.

R3_17:

ED Fig 2: maybe state that this was initial conditions?

This particular figure (Extended Data Fig. 2 in original manuscript; now Supplementary Fig. 3) has data for both initial and treatments (see x-axis labels).

REVIEWERS' COMMENTS

Reviewer #1 (Remarks to the Author):

Manganese co-limitation of phytoplankton growth and major nutrient drawdown in the Southern Ocean

Browning et al.

In the revised version of the manuscript entitled "Manganese co-limitation of phytoplankton growth and major nutrient drawdown in the Southern Ocean", the authors have addressed most of my comments and suggestion. The introduction reads well, properly setting the bases about what is being discussed in the manuscript, and the discussion section has been improved interconnecting the main points of this study.

The QA-QC data and the description of the analytical procedures added to the method section, as well as the re-writing of the modelling section (adding sensitivity analysis and a much more comprehensive explanation of the model and its implications) greatly improve the manuscript reading.

That being said, I would like to see a brief summary about the circulation and key water masses present in the study area. This will help readers, not familiarized with the Southern Ocean, to understand the processes described in the manuscript such as offshore shelf transport of DMn and relative influence of deep waters discussed in the text.

Specific Comments:

Line 205: Remove the repeated "at": ... nutrient supply at either Fe or Mn limited sites.

Line 677: Connected to my comment about the circulation in this regions, please provide, if available, references for offshore transport time.

Line 1079: Are the labels correct in this Figure; it is stated that "d-e, mean observations for shelf-impacted (circles in 'a'), and f-g, open ocean waters (crosses in 'a')". However, in line 431 the open ocean scenario is referred as fig 3e.

Reviewer #2 (Remarks to the Author):

I have read the revised submission of "Manganese co-limitation of phytoplankton growth and major nutrient drawdown in the Southern Ocean" by Browning et al. and considered their response to my original review. To address my concerns, the authors have overhauled the modeling section, which is now much easier to follow. The emphasis on short timescales (<20 years) makes the conclusions in Fig. 3 more robust (although I also thank the authors for graciously correcting my misconceptions about the van Hulst et al. (2017) model).

My only major comment is that the paragraph at Line 233 seems to end without summarizing the results of the ecosystem simulations in d/e/f/g. Does the model predict phytoplankton Mn limitation in addition to low Mn*? Regardless, a succinct sentence summarizing the utility of the modeling with respect to the Mn* and/or incubation results might be warranted. Is the model mostly used to show how Mn* will evolve over time by scavenging kinetics, or is it backing up the observations that Mn limitation will emerge when $Mn^* < 0$?

Overall, I fully support publication of this work. In my re-reading I have a few small comments and line edits that the authors may wish to consider, but should not be bound to change.

26: clarify that this is oxygenic photosynthesis?

30: "typically substantially enriched" could be clearer

120: I had a hard time with the last half of this paragraph and I suspect others might not grasp its meaning clearly either. Nutrient ratios are challenging because the ratios of standing stocks are the inverse of phytoplankton uptake ratios and so one is forced to mentally flip back and forth to consider what increases and decreases actually mean. Improving the paragraph's final sentence to summarize the preceding arguments might fix this easily.

Line 132: does it make more sense to have this paragraph earlier in this section given the fact that community composition has already been invoked at Lines 95 and 122?

Line 274: Similar to Line 120, it was hard to follow the directionality of these arguments. Simplifying these sentences (e.g. Line 283 removing "rather than Fe limitation" makes the subject much more clear) would help.

Line 437 and afterwards: Perhaps this narrative should be moved to the supplement?

Fig 3. Is the legend reversed for d+e and f+g? My assumption based on the concentrations in panel G is that this was initialized as a coastal site but lines 799-800 suggest this is actually d+e?

Reviewer #3 (Remarks to the Author):

I found this version of the paper significantly improved, and the modelling study much better integrated and the scope of the modelling much more in line with the papers goals. My main concerns about the modeling – the nature of the "deep water", and the vagueness of initial conditions of the short term experiments have been dealt with.

I did really struggle with Fig 3d-g. Is the caption wrong? Are d and e for open ocean and f and g for shelf. If this is a mistake things make sense, but if the caption is correct I obviously have really not understood the paper. As it is, the authors do not really talk through these results enough i.e. lead the author through what features of these plots are important. Some can be done in methods, but at least given enough info (would have helped that I knew which plots really were for which place). Things like the larger seasonal drawdown in iron is an interesting feature in g etc will be useful to mention and explain.

We would assume Mn limitation if Mn* is negative – but in fact in (e) it is very low, but not negative (this is for (c) as well. I suggest some explanation about this.

Line 115-116: I found this rather opaque, suggest expanding

Line 154: $R_{fe:m}$ is a mean. In response to my previous query of this as a constant the authors respond that as long as it is the same at all sites (and I assume they also mean time) it won't impact the qualitative nature of the results. But it likely does change between the site and with season, and if it changes an order of magnitude it really could alter the results – so I'm not so sure that it isn't an issue. Maybe at least add a caveat to this effect.

Line 168-172: I think there is something grammatically wrong with this sentence

Line 193: why use just data from the Antarctic Peninsula (i.e. why not near shore on north side of Drake Passage)? Is this all that was available?

Line 219-220: this is a bit vague. Makes some sense having read the responses to reviewers, but suspect to other readers this needs a bit more explanation

Line 223-224: similarly I suggest a bit more detail about why there is photoreduction of Mn would help a less expert reader understand the rest of this paragraph

Line 393: why ligand=0.8nM? The authors explore lower (0.6nM) but not higher. Though GEOTRACERS have found a much larger range of ligand concentrations.

Does the first set of ecosystem have dust (the text seems to imply not). Why not? Is the assumption that dust supplies are too low to matter? nat

Response to Reviewer comments

We thank the three Reviewers for their positive evaluation of our earlier revision. Responses to their follow up comments are provided below in blue.

Reviewer #1 (Remarks to the Author):

Manganese co-limitation of phytoplankton growth and major nutrient drawdown in the Southern Ocean

Browning et al.

In the revised version of the manuscript entitled “Manganese co-limitation of phytoplankton growth and major nutrient drawdown in the Southern Ocean”, the authors have addressed most of my comments and suggestion. The introduction reads well, properly setting the bases about what is being discussed in the manuscript, and the discussion section has been improved interconnecting the main points of this study.

The QA-QC data and the description of the analytical procedures added to the method section, as well as the re-writing of the modelling section (adding sensitivity analysis and a much more comprehensive explanation of the model and its implications) greatly improve the manuscript reading.

That being said, I would like to see a brief summary about the circulation and key water masses present in the study area. This will help readers, not familiarized with the Southern Ocean, to understand the processes described in the manuscript such as offshore shelf transport of DMn and relative influence of deep waters discussed in the text.

We thank the Reviewer for their supportive comments and additional suggestions.

We have now conducted some brief revisions to note the current systems and relationship to off-shelf transport (with supporting references), in the context of the model simulations (in the model section; lines 234–235 of the track changes version). Deep water upwelling is noted in lines 195–196 of the track changes version.

Specific Comments:

Line 205: Remove the repeated “at”: ... nutrient supply at either Fe or Mn limited sites.

Corrected

Line 677: Connected to my comment about the circulation in this regions, please provide, if available, references for offshore transport time.

A note on timescales for off shelf transport (days–weeks), alongside supporting references, has now been included here.

Line 1079: Are the labels correct in this Figure; it is stated that “d–e, mean observations for shelf-impacted (circles in ‘a’), and f–g, open ocean waters (crosses in ‘a’)”. However, in line 431 the open ocean scenario is referred as fig 3e.

There was an error in the Figure 3 legend: the incorrect panels were referred to. This has

been corrected in the revised manuscript.

Reviewer #2 (Remarks to the Author):

I have read the revised submission of “Manganese co-limitation of phytoplankton growth and major nutrient drawdown in the Southern Ocean” by Browning et al. and considered their response to my original review. To address my concerns, the authors have overhauled the modeling section, which is now much easier to follow. The emphasis on short timescales (<20 years) makes the conclusions in Fig. 3 more robust (although I also thank the authors for graciously correcting my misconceptions about the van Hulst et al. (2017) model).

My only major comment is that the paragraph at Line 233 seems to end without summarizing the results of the ecosystem simulations in d/e/f/g. Does the model predict phytoplankton Mn limitation in addition to low Mn*? Regardless, a succinct sentence summarizing the utility of the modeling with respect to the Mn* and/or incubation results might be warranted. Is the model mostly used to show how Mn* will evolve over time by scavenging kinetics, or is it backing up the observations that Mn limitation will emerge when Mn* < 0?

We thank the Reviewer for their supportive comments and additional suggestions.

Indeed, the model was used to investigate the relative influence of surface mixed layer Fe scavenging, Mn redox processes, wintertime nutrient entrainment, and spring-summer phytoplankton growth on changes in Mn* and the resultant identity of the nutrient limiting phytoplankton growth.

We have now ended this paragraph with two sentences summarizing ecosystem model findings. Following on from our existing statement that changes in photoreduction rates throughout the year have less impact on DFe and DMn than wintertime entrainment and springtime phytoplankton growth, we summarize with the following:

“As a result, values of surface ocean Mn in the model remained elevated (Fe deficient) in the near shelf case and close to 0 nM (Mn-Fe co-deficient) in the open ocean case, reflecting the sub-mixed layer Fe and Mn supply stoichiometry. Spring-summer phytoplankton growth in the model therefore becomes Fe-Mn co-limited in the open ocean case (Fig. 3d,e) and Fe limited in the near shelf case (Fig. 3f,g), matching our bioassay experiment results (Figs 1 and 2).”*

Overall, I fully support publication of this work. In my re-reading I have a few small comments and line edits that the authors may wish to consider, but should not be bound to change.

26: clarify that this is oxygenic photosynthesis?

Now clarified as oxygenic photosynthesis.

30: “typically substantially enriched” could be clearer

Sentence now rephrased to: *“Throughout much of the global ocean, dissolved Mn concentrations are characterized by surface ocean maxima and depletion in the ocean interior (typically <0.2 nM)^{3,4}.”*

120: I had a hard time with the last half of this paragraph and I suspect others might not grasp its meaning clearly either. Nutrient ratios are challenging because the ratios of standing stocks are the inverse of phytoplankton uptake ratios and so one is forced to mentally flip back and forth to consider what increases and decreases actually mean. Improving the paragraph's final sentence to summarize the preceding arguments might fix this easily.

We have now ended this paragraph with a simple summary sentence as a take home message for readers: *"In summary, prevalence of Mn versus Fe limitation appeared important for the relative drawdown of Southern Ocean silicic acid and nitrate following micronutrient supply."*

Line 132: does it make more sense to have this paragraph earlier in this section given the fact that community composition has already been invoked at Lines 95 and 122?

We agree and have now shifted the paragraph describing phytoplankton community structure by one position. We also note that as a result of this, the order of Supplementary Figures 2, 3, and 4 has also been altered.

Line 274: Similar to Line 120, it was hard to follow the directionality of these arguments. Simplifying these sentences (e.g. Line 283 removing "rather than Fe limitation" makes the subject much more clear) would help.

Minor amendments to wording/sentence structure have been made to this paragraph to improve clarity.

Line 437 and afterwards: Perhaps this narrative should be moved to the supplement?

As there are some important methods descriptions included here (e.g., description of the ecosystem model), we have decided to leave this section in the main manuscript Methods. We are however happy to move this to a Supplementary Methods or Supplementary Note, should the Editor request this.

Fig 3. Is the legend reversed for d+e and f+g? My assumption based on the concentrations in panel G is that this was initialized as a coastal site but lines 799-800 suggest this is actually d+e?

There was an error in the Figure 3 legend: the incorrect panels were referred to. This has been corrected in the revised manuscript.

Reviewer #3 (Remarks to the Author):

I found this version of the paper significantly improved, and the modelling study much better integrated and the scope of the modelling much more in line with the papers goals. My main concerns about the modeling – the nature of the "deep water", and the vagueness of initial conditions of the short term experiments have been dealt with.

I did really struggle with Fig 3d-g. Is the caption wrong? Are d and e for open ocean and f and g for shelf. If this is a mistake things make sense, but if the caption is correct I obviously have really not understood the paper. As it is, the authors do not really talk through these results enough i.e. lead the author through what features of these plots are important. Some

can be done in methods, but at least given enough info (would have helped that I knew which plots really were for which place). Things like the larger seasonal drawdown in iron is an interesting feature in g etc will be useful to mention and explain.

We would assume Mn limitation if Mn^* is negative – but in fact in (e) it is very low, but not negative (this is for (c) as well. I suggest some explanation about this.

We thank the Reviewer for their supportive comments on our previous revision and additional suggestions.

There was an error in the Figure 3 legend: as the reviewer suspected, the incorrect panels were referred to. We apologize for the confusion this caused. This has been corrected in the revised manuscript.

As the reviewer notes, the Mn^* in the open ocean ecosystem model simulation is indeed positive, but close to 0 nM. This is a result of the average open ocean, deep-water GEOTRACES Mn^* , which is used to parameterize deep waters in the model, also being slightly positive (although there are many individual negative values—see red dashes in Fig. 3c—we objectively use the mean). Because of the importance of wintertime entrainment in this Southern Ocean setting for regulating mixed layer DFe and DMn concentrations (relative to e.g., Mn photoreduction), this largely sets the available DFe:DMn to phytoplankton in the model surface layer, leading to their Fe-Mn co-limited growth.

In the revised manuscript we have now made small adjustments to the paragraph describing the model and added two sentences to summarize model changes in Mn^* and phytoplankton nutrient limitation under the open ocean and near shelf scenarios.

Line 115-116: I found this rather opaque, suggest expanding

This sentence has been modified to improve clarity. We point out that further explanations follow this initial statement (to which we have also made some small additional changes in the revised manuscript to improve clarity).

Line 154: $R_{Fe:Mn}$ is a mean. In response to my previous query of this as a constant the authors respond that as long as it is the same at all sites (and I assume they also mean time) it won't impact the qualitative nature of the results. But it likely does change between the site and with season, and if it changes an order of magnitude it really could alter the results – so I'm not so sure that it isn't an issue. Maybe at least add a caveat to this effect.

We agree with the Reviewers point and have included a caveat to the revised manuscript, noting the assumption of a conserved phytoplankton Fe:Mn stoichiometry:

“Provided the assumed-average Fe:Mn ratio of phytoplankton, $R_{Fe:Mn}$, remains relatively conserved across the spatial-temporal timescales of the observations, our experimental results suggest that these low Mn^ values also reflect widespread Mn limitation, or Fe-Mn co-limitation, throughout open Southern Ocean regions.”*

Line 168-172: I think there is something grammatically wrong with this sentence

This sentence has now been rephrased.

Line 193: why use just data from the Antarctic Peninsula (i.e. why not near shore on north side of Drake Passage)? Is this all that was available?

This was indeed the extent of the data availability in the region.

Line 219-220: this is a bit vague. Makes some sense having read the responses to reviewers, but suspect to other readers this needs a bit more explanation

This sentence has now been revised:

“The low Mn of these isolated deep waters prime them for phytoplankton Mn limitation, or Mn-Fe co-limitation, upon transfer to the surface ocean¹⁰, provided they do not interact substantially with high Mn* shelf-impacted waters previously described.”*

Line 223-224: similarly I suggest a bit more detail about why there is photoreduction of Mn would help a less expert reader understand the rest of this paragraph

We have now clarified this statement in the revised manuscript:

“Once entrained into surface mixed layers, the difference in removal rate between DFe and DMn in both scenarios is extended by photoreduction of Mn, which can speed up conversion of Mn oxides to free Mn ions by >50 times compared to rates in the dark^{6,7}. This both protects DMn from removal and dissolves Mn oxides entrained from deep waters^{6,7} and is a key mechanism maintaining elevated DMn in surface waters of the lower latitude oceans^{3,4,8}.”

Line 393: why ligand=0.8nM? The authors explore lower (0.6nM) but not higher. Though GEOTRACERS have found a much larger range of ligand concentrations.

Does the first set of ecosystem have dust (the text seems to imply not). Why not? Is the assumption that dust supplies are too low to matter?

Running the simulations with higher ligand concentrations, e.g., 1 nM, results in DFe concentrations plateauing at slightly higher concentrations than for the 0.8 nM simulation and leads to little change in simulated Mn*. This additional sensitivity test result (with 1 nM ligands) has been added to the Methods.

The first set of ecosystem simulations has no dust supply. This makes them easier to understand in terms of Mn redox processes, Fe scavenging, wintertime deep-water entrainment, and biological uptake. The impact of adding a modern-day dust supply is shown in Supplementary Figure 6 (upper panels). As this latter simulation shows, a modern-day dust supply is sufficiently low that changes are not easily distinguished from a ‘no dust’ scenario (i.e., only slightly decreasing Mn* relative to no dust deposition at all). This is clarified by a minor addition to the text that compares the glacial high dust simulation with the modern-day low dust one.